# Draft genome assembly and transcriptome data of the icefish *Chionodraco myersi* reveal the key role of mitochondria for a life without hemoglobin at subzero temperatures

Luca Bargelloni [1,2,3]*, Massimiliano Babbucci[1], Serena Ferraresso[1], Chiara Papetti[3,4], Nicola Vitulo [5], Roberta Carraro[1], Marianna Pauletto [1], Gianfranco Santovito[3], Magnus Lucassen[6], Felix Christopher Mark [6], Lorenzo Zane [2,3] & Tomaso Patarnello[1,3]

Antarctic fish belonging to Notothenioidei represent an extraordinary example of radiation in the cold. In addition to the absence of hemoglobin, icefish show a number of other striking peculiarities including large-diameter blood vessels, high vascular densities, mitochondria-rich muscle cells, and unusual mitochondrial architecture. In order to investigate the bases of icefish adaptation to the extreme Southern Ocean conditions we sequenced the complete genome of the icefish *Chionodraco myersi*. Comparative analyses of the icefish genome with those of other teleost species, including two additional white-blooded and five red-blooded notothenioids, provided a new perspective on the evolutionary loss of globin genes. Muscle transcriptome comparative analyses against red-blooded notothenioids as well as temperate fish revealed the peculiar regulation of genes involved in mitochondrial function in icefish. Gene duplication and promoter sequence divergence were identified as genome-wide patterns that likely contributed to the broad transcriptional program underlying the unique features of icefish mitochondria.

[1] Department of Comparative Biomedicine and Food Science, University of Padova, Viale dell'Università 16, 35020 Legnaro, Italy. [2] Department of Land, Environment, Agriculture, and Forestry, University of Padova, Viale dell'Università 16, 35020 Legnaro, Italy. [3] Consorzio Nazionale Interuniversitario per le Scienze del Mare (CoNISMa), Piazzale Flaminio 9, 00196 Rome, Italy. [4] Department of Biology, University of Padova, Via G. Colombo 3, 35131 Padua, Italy. [5] Department of Biotechnology, University of Verona, Strada Le Grazie 15, 37134 Verona, Italy. [6] Section of Integrative Ecophysiology, Alfred Wegener Institute Helmholtz. Centre for Polar and Marine Research, Am Handelshafen 12, Bremerhaven 27570, Germany. *email: luca.bargelloni@unipd.it

Antarctic fish belonging to the family Channichthyidae are the only vertebrates that lack hemoglobin as first reported by Ruud[1]. They were named icefish by the British whalers sailing the Southern Ocean because of their translucent aspect, due to the absence of respiratory pigments and scaleless skin. The icefish are part of a broader taxonomic group, the Notothenioidei, which dominate the Antarctic fish fauna[2]. Notothenioids represent an extraordinary example of radiation in the cold, as they have filled several ecological niches emptied by the dramatic cooling of the Southern Ocean in the last 25 myr[3,4]. The extremely low water temperature (−1.8 °C) is also considered the necessary condition for the evolutionary loss of hemoglobin in the icefish. According to Henry's law[5], oxygen solubility is inversely correlated with water temperature, therefore in the subzero Southern Ocean the concentration of dissolved oxygen is substantially higher than in temperate waters. Such a condition, however, is not sufficient to compensate for the lack of hemoglobin. In fact, the icefish shows several cardiovascular modifications (e.g., greater blood volume, wider capillaries, and larger hearts) to offset the absence of oxygen carriers (reviewed by Sidell and O'Brien[6]). At the cellular level, it has long been reported that icefish show extraordinarily high mitochondrial density[7,8]. An increased ratio between mitochondrial and muscle cell volume appears to be a more general response to cold-acclimation in fish[9–11]. The Antarctic notothenioids and, in particular, the hemoglobinless icefish represent an extreme example of such response[12]. The role of mitochondria in cold-acclimation of ectothermic vertebrates is rather complex and remains to be fully elucidated. Going beyond the classic hypothesis that higher mitochondrial density provides compensatory increase of aerobic capacity during cold-acclimation, Sidell[13] suggested that denser mitochondrial membranes might favor intracellular oxygen transport. Oxygen diffusion, which is reduced in aqueous solutions at low temperatures, is greatly enhanced in lipid bilayers. In the icefish, where oxygen transport cannot be facilitated by hemoglobin, mitochondrial lipid membranes might be the key to ensure its efficient diffusion and storage within the cell. In fact, the white-blooded channichthyids have higher phospholipid mitochondrial content and smaller surface-to-volume ratio compared to red-blooded notothenioids[14], which suggests a larger degree of mitochondrial fusion relative to fission[15]. The large icefish mitochondria, however, show sparser mitochondrial cristae, which are the mitochondrial substructures where respiratory complexes are located[16]. This is in keeping with the evidence that channichthyids do not show a proportionally higher aerobic metabolic capacity[8,17]. In recent years, there has been a resurging interest in mitochondrial biology, largely due to the appreciation of the role of mitochondria in multiple functions in addition to ATP production[18]. Studies on mitochondrial dynamics[16] and their interactions with other cellular organelles[19] reveal an unsuspected complexity and provided clues to explore the role of mitochondria in the adaptation to life without hemoglobin.

The evolution of the unique hemoglobinless condition of icefish has understandably attracted much attention and several studies investigated the molecular basis of hemoglobin loss and addressed the question whether it might be a form of adaptation to the extreme Antarctic environment or, conversely, it represents an example of disaptation[6,20]. Cocca et al.[21,22] characterized the remnants of the globin locus. They found that the same partial copy of one alpha-globin gene was present in several white-blooded species, differently from beta-globin genes, which were not detected. They hypothesized that a single event of deletion in the common ancestor of channichthyids might have been the driver of the evolution of the hemoglobinless condition. A subsequent study[23] discovered that a highly derived channichthyid species, Neopagetopsis ionah, shows a complete copy of both alpha- and beta-globin genes, calling into question the hypothesis of a single event for the evolutionary loss of hemoglobin. Both studies, however, assumed a single-genomic locus for alpha- and beta-globins. Genomic analysis showed that two genomic globin clusters are present in the teleost genome, with multiple copies of globin genes within each cluster[24], prompting for whole-genome investigations on globin gene evolution in the Notothenioidei. In parallel to the complete absence of hemoglobin, icefish show very few circulating erythrocytes. Comparing the transcriptome of primary erythropoietic tissues between a white-blooded species, Chionodraco hamatus, and two red-blooded notothenioids, Xu et al.[25] demonstrated the silencing of several key transcription factors involved in erythropoiesis, possibly mediated by upregulation of suppressive microRNAs. A second transcriptome study reported that microRNAs required for erythropoiesis are conserved in white-blooded notothenioids[26]. At variance with the nearly complete deletion of globin genes, these studies suggest that downregulation rather than gene loss underlies the limited presence of erythrocytes in icefish blood, although confirmation at the genome level is still lacking.

A decade ago, Chen et al.[27] discovered a general pattern in the evolution of the notothenioid genome. Using EST data and microarray-based comparative genomic hybridization they showed that massive gene duplication led to the dramatic expansion of several gene families. Such genomic evolutionary pattern was later demonstrated to provide the basis for the neofunctionalization of zona pellucida (ZP)-like proteins, which are crucial for freeze prevention of notothenioid eggs[28]. Similarly, the analysis of the muscle transcriptome of the icefish C. hamatus discovered selective duplication of genes encoding mitochondrial proteins compared to other teleost species[29]. The role of gene duplication in Antarctic fish evolution has been further confirmed after the recent publication of the draft genomes of two red-blooded notothenioids, Eleginops maclovinus and Dissostichus mawsoni[30] and one white-blooded species, Chaenocephalus aceratus[31]. Such genomic information also provided us with the unprecedented opportunity for a comparative analysis at the genome and transcriptome level between red- and white-blooded notothenioids. Such comparative analysis included E. maclovinus, which the closest outgroup of all Antarctic notohenioids and it belongs to a lineage that never inhabited the waters surrounding Antarctica and P. charcoti, which belongs to the family Bathydraconidae, which represents the closest living relatives to the hemoglobinless channichthyids[32,33]. Here, we report the draft genome assembly and muscle transcriptome data for a second channichthyid species, C. myersi, and we identify the genomic and transcriptomic patterns underlying the key role of mitochondria in the evolution of this unique group of vertebrates.

## Results

**Icefish genome assembly and annotation**. The draft genome of C. myersi was assembled into 63,605 scaffolds for a total size of 1.12 Gbp and contained 38,127 putative protein-coding genes. The estimated genome size is very similar to that reported for another icefish (1.1 Gbp, C. aceratus)[31] and larger than those of two red-blooded notothenioid species (0.73 Gbp, E. maclovinus; 0.84 Gbp D. mawsoni)[30]. For all these species, genome size based on k-mer analysis is lower than estimated using DNA content and BAC library sequencing by Detrich et al.[34]. It is difficult to explain the reason of such a discrepancy, although a likely hypothesis is that the high content in repetitive sequences, especially for white-blooded notothenioids[34] might lead to underestimating genome size when using next-generation sequence data. Benchmarking Universal Single-Copy Orthologs (BUSCO) analysis showed good completeness (86.8%) and

representation (93.5%). The same analysis was run for four more nothenioid genomes (*Nothenia coriiceps, C. aceratus, E. maclovinus, D. mawsoni;* Supplementary Table 1) showing that the level of completeness and representation for *C. myersi* is within the range observed for these species (80–97%). All details about *C. myersi* genome sequencing, assembly, k-mer analysis, and annotation are reported in Supplementary Methods, in Supplementary Tables 2–4 and in Supplementary Fig. 1. Putative protein-coding sequences for *C. myersi* were analyzed together with 18 other teleost genomes, including other nothenioid species, and 21,718 orthology groups (OGs) were identified (Supplementary Table 5).

**Whole-genome analysis of globin gene loss and the evolution of the erythropoiesis pathway.** Conserved flanking orthologs were used as anchors to identify the contiguous genomic regions corresponding to the two teleost globin gene clusters (MN and LA as defined in Opazo et al.[24]) in the hemoglobinless icefish *C. myersi* and *C. hamatus.* For this purpose, a draft genome of *C. hamatus* was assembled (see Supplementary Methods section). Kim et al.[31] recently compared the genomic organization of the LA and MN clusters between the icefish *C. aceratus* and temperate fish species. Here, we included in the comparison seven nothenioid species[30,31,35–37], namely three white-blooded icefish and four red-blooded nothenioids, *N. coriiceps, P. charcoti, D. mawsoni,* and *E. maclovinus* (Fig. 1a). *Eleginops maclovinus* is particularly interesting as it is the closest outgroup of all Antarctic nothenioids and it belongs to a lineage that never inhabited the waters surrounding Antarctica, while *P. charcoti* belongs to the family Bathydraconidae, which represents the closest living relatives to the hemoglobinless channichthyids[32,33].

In the LA cluster only one alpha-globin fragment was found in the three icefish genomes, as previously reported[22], while cluster MN was precisely defined by orthologous anchors, but showed no trace of globin-like sequences as already described[31]. The same analysis carried out on the genomes of the four red-blooded nothenioids was extended to the sub-Antarctic species *E. maclovinus* that showed the presence of up to 10 full-length genes encoding beta-and alpha-globins (Fig. 1a).

Analysis of sequence similarity between the MN globin genomic cluster in *E. maclovinus* and the homologous region in *C. myersi* and *C. aceratus* showed at least two conserved sequence fragments (indicated by arrows in Fig. 1b). Such evidence suggests that multiple events of deletion and/or rapid sequence divergence, in addition to the already reported deletion in the LA cluster, led to the loss of globin genes. Unlike the fate of the two globin clusters, full-length key erythropoietic transcription factors were identified (full list in Table S2 of Xu et al.[25]) in the *C. myersi* genome, but were found to be suppressed in the icefish. To further explore the role of selection on proteins involved in erythropoiesis, we used RELAX, a method specifically developed to identify relaxation of purifying selection on protein-coding genes[39]. Seven strictly one-to-one orthologs encoding important erythropoietic factors (GFI1B, RHAG, TAL1, LMO2, ALAD, CPOX, PPOX) were analyzed across six teleost fish (*Danio rerio, Oreochromis niloticus, Gasterosteous aculeatus, E. maclovinus, D. mawsoni, C. myersi*) to test whether in the icefish evolutionary lineage relaxed purifying selection could be observed. Although five out of seven genes suggested relaxation of evolutionary constraints, none of these tests were statistically significant (Supplementary Table 6).

**Comparative transcriptomic analysis of icefish muscle expression profiles.** Transcriptome data of *Chionodraco myersi* skeletal muscle from five individual samples were generated and analyzed

with comparable data (adult muscle tissue, five biological replicates) that were available in public repositories for zebrafish (*D. rerio*), Nile tilapia (*O. niloticus*), and threespine stickleback (*G. aculeatus*). These three species all have high quality, well-annotated genomes and represent diverse teleost evolutionary lineages adapted to different thermal regimes. Since divergence at the transcriptome level between model fish species and icefish might be due either to the peculiar hemoglobinless condition or to the response to subzero temperatures, we included in the analysis muscle RNA-seq data from Antarctic red-blooded nothenioids, which share with icefish the long-term adaptation to the Southern Ocean conditions, but still retain functional hemoglobins. Unfortunately, a sufficient number of biological replicates was not available for a single nothenioid species in public repositories. Therefore, we decided to group together RNA-seq data from three red-blooded Antarctic species (*D. mawsoni, N. coriiceps, P. charcoti,* see methods). Although such an approach might inflate expression variance within this group and be overly conservative, we considered that it was crucial to compare at a transcriptome-wide level, the hemoglobinless icefish and the red-blood nothenioids. After quality filtering and normalization, RNA-seq expression data were compared for 9721 OGs by hierarchical clustering and principal component analyses (Fig. 2). Results showed a clear separation between *C. myersi* and all the other species, including red-blooded nothenioids, which were grouped together despite representing cross-species data. Differential expression was statistically assessed between *Chionodraco* and all three model species together and similar results were obtained when using only red-blooded nothenioid. Red- and white-blooded nothenioid expression data were analyzed separately (see methods).

Comparing *C. myersi* against the three model species, differential expression analysis identified 2758 and 2813 OGs over- and under-expressed, respectively (logFoldChange (logFC) $\geq 1$ or logFC $\leq -1$, false-discovery rate (FDR) $\leq 0.05$). The comparison between white- and red-blooded nothenioids yielded similar results, with 1630 up- and 2882 downregulated OGs. Gene-set enrichment analysis (GSEA) was then employed to specifically test the hypothesis that mitochondria play a key role in compensating for the lack of hemoglobin. In the comparison between icefish and non-Antarctic species, two gene sets, IMPI and Mitocarta, which include, respectively, 1626 and 1158 proteins with mitochondrial localization, were enriched with high significance (IMPI FDR *q*-val 0, normalized enrichment score (NES) 3.9; MitoCarta FDR *q*-val 0, NES 3.6). Likewise, two gene sets that contain genes predicted to be under the control of two key regulators of mitochondrial biogenesis, NRF1 and NRF2/GABPA, were significantly enriched (NRF1 FDR *q*-val 0.003, NES 2.3; NRF2/GABPA, FDR *q*-val 0.1, NES 1.7). Even stronger evidence for differential expression of mitochondria-related genes was observed in the comparison between white- and red-blooded nothenioids (IMPI FDR *q*-val 0, NES 3.4; MitoCarta FDR *q*-val 0, NES 3.2, NRF1 FDR *q*-val 0, NES 1.9; NRF2/GABPA, FDR *q*-val 0.003, NES 1.8). Two additional gene sets were significantly enriched, MITOCHONDRIAL_INNER_MEMBRANE (FDR *q*-val 0.006, NES 1.9) and BIOCARTA_PGC1A_PATHWAY (FDR *q*-val 0.03, NES 1.9). The inner mitochondrial membrane hosts the electron transport chain machinery, generates membrane potential necessary for ATP generation, and forms the signature folds of mitochondria, known as cristae[40]. Genes in the PCG-1α pathway are crucial for mitochondrial biogenesis[41].

Detailed analysis of genes involved in various mitochondrial processes confirmed evidence of differential expression in the icefish as compared to model species. Among these, it is worth to note that the large majority (8–10 out of 13, Table 1) of genes encoding cofactors involved in mitoribosome assembly were

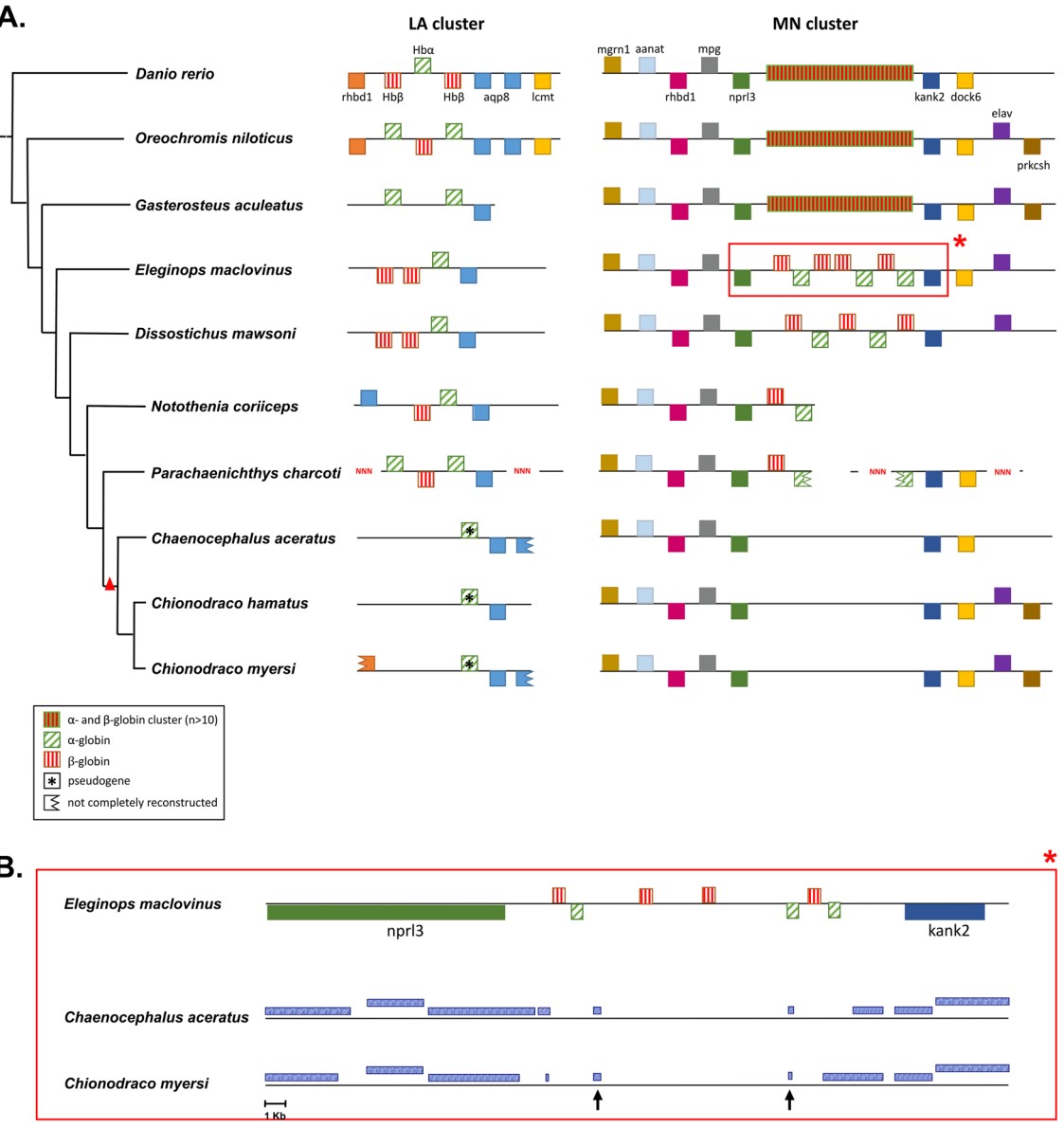

**Fig. 1** LA and MN globin clusters genomic structure. **a** Genomic organization (not to scale) of the LA and MN globin gene clusters of teleost fish. The phylogenetic tree depicted in the figure is based on[32,33,38]. All clusters are presented in the same orientation as *D. rerio*. Genes in forward orientation are shown on top of the contig whereas genes in the reverse orientation are shown below. Red triangle depicts loss of functional hemoglobins. Hbβ = beta-globin, Hbα = alpha-globin, aqp8 = aquaporin 8, lcmt = leucine carboxyl methyltransferase, mgrn1 = mahogunin ring finger 1, aanat = arylalkylamine N-acetyltransferase, rhbd1 = inactive rhomboid protein 1, mpg = DNA-3-methyladenine glycosylase, nprl3 = nitrogen permease regulator 3-like, kank2 = KN motif and ankyrin repeat domain-containing protein 2, dock6 = dedicator of cytokinesis protein 6, prkcsh = glucosidase II subunit beta. The asterisk refers to the genomic region magnified in Fig. 1b. **b** Sequence conservation of the genomic region flanking MN globin cluster either between *E. maclovinus* and *C. aceratus* or between *E. maclovinus* and *C. myersi*. Blue boxes represent Blastn best hits. The two genes, nprt3 and kank2, appear to be fully functional in both icefish genomes. Arrows in Fig. 1b indicate regions with sequence similarity between *E. maclovinus* and either *C. aceratus* or *C. myersi* in the genomic region where globin genes are present in red-blooded notothenioids.

over-expressed. Assembly of the large and small ribosomal subunits is essential for mitochondrial biogenesis[42]. Also, proteins that play a fundamental role in mitochondrial fusion/fission and cristae remodeling[16] were differentially regulated at the messenger RNA (mRNA) level (Table 2). Transcripts encoding three proteins forming the mitochondrial contact site and cristae organizing system (MICOS), including IMMT (MIC60), a core MICOS component[43], were significantly under-expressed (FDR < 0.05) in *C. myersi*. Mitochondrial creatine-kinase (mtCK), a relevant protein for bridging inner and outer mitochondrial membranes and promoting lipid transfer (albeit its role is still controversial)[27,29–31], is largely under-expressed (Table 2).

Interestingly, mtCK knock-out in mice produces a muscle cellular phenotype similar to that of icefish, with an ultra-structural remodeling of muscle fibers, leading to higher mitochondrial density and enlarged mitochondria[44]. It was suggested that lack (or very low levels) of mtCK could be compensated by high levels of adenylate kinase (AK)[45]. This indeed seems to be the case in *C. myersi*, which exhibits significant (FDR < 0.05) upregulation of several AK genes such as AK1, AK2, AK4, and AK9 (Table 2). Also OPA1, a fundamental player in cristae remodeling, was downregulated[46]. Finally, the key regulator of mitochondrial fission, DNM1L/DRP1, was downregulated, whereas Mitoguarding (MIGA1), which favors mitochondrial fusion[47] and tethers

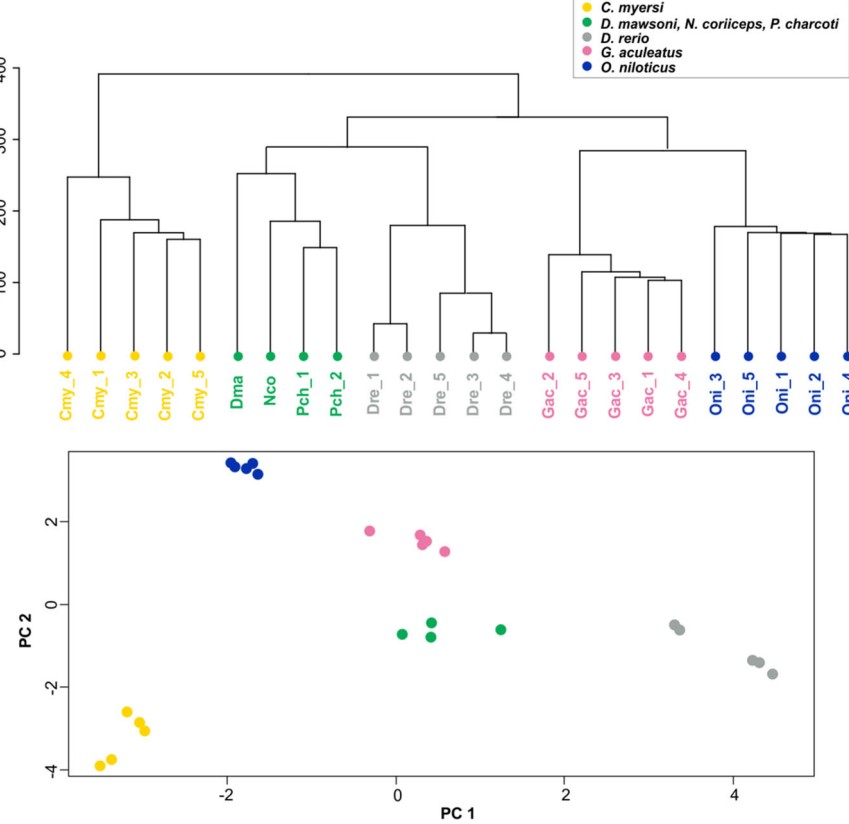

**Fig. 2** Hierarchical clustering and principal component analyses of muscle RNA-seq data. Clustering and spatial distribution of muscle RNA-seq data. Cmy: *C. myersi*; Dre: *D. rerio*; Gac: *G. aculeatus*; Oni: *O. niloticus*, Dma: *D. mawsoni*; Nco: *N. coriiceps*; Pch: *P. charcoti*. Biological replicates are distinguished by numbers from 1 to 5.

**Table 1 Muscle differential expression analysis of genes encoding cofactors involved in mitoribosome assembly.**

| Gene symbols | OG ID | Cmy vs. Red | | Cmy vs. NonAnt | |
|---|---|---|---|---|---|
| | | LogFC | FDR | LogFC | FDR |
| MRM1 | OG0012273 | 8.70 | 1.08E-12 | 2.53 | 0.00001 |
| MRM2 | OG0013246 | 0.46 | 0.68674 | 1.79 | 0.01748 |
| MRM3 | OG0007072 | 1.49 | 0.00828 | 3.56 | 1.77E-19 |
| MTG2 | OG0008966 | 1.51 | 0.00453 | 2.53 | 5.52E-11 |
| DDX28 | OG0008112 | −0.16 | 0.83706 | 1.15 | 0.01680 |
| DHX30 | OG0012049 | −0.78 | 0.24656 | 1.43 | 0.00214 |
| MTERF3 | OG0012260 | 1.09 | 0.02093 | 1.70 | 3.81E-07 |
| AFG3L2 | OG0001029 | −0.50 | 0.29814 | −0.50 | 0.16280 |
| SPG7 | OG0011454 | 0.93 | 0.10783 | 0.92 | 0.01888 |
| TFB1M | OG0005933 | 2.58 | 0.00007 | 3.58 | 7.52E-18 |
| NSUN4 | OG0011353 | −1.15 | 0.15217 | −0.53 | 0.43104 |
| ERAL1 | OG0009078 | 0.40 | 0.46238 | 1.56 | 2.9E-06 |
| MPV17L2 | OG0006317 | −0.87 | 0.30116 | −0.68 | 0.32989 |

*Cmy Chionodraco myersi, Red Red-Blooded Antarctic fish species, NonAnt non-Antarctic species*

mitochondria to lipid droplets (L. Scorrano, personal communication), was upregulated (Table 2).

In consideration of the potential role of mitochondrial lipids in the adaptation to low temperature and loss of hemoglobin, we then examined genes involved in lipid synthesis. Several genes encoding key enzymes in glycerolipids biosynthetic pathways (GPAT1/2, GPAT3/4, LPAAT, AGPAT2) were significantly upregulated (FDR < 0.05) in *Chionodraco* compared to non-Antarctic species. Increased expression of GPAT3/4 and LPAAT might favor phosphatidic acid (PA) synthesis. PCYT1A, which is important for regulating phosphatidylcholine (PC) biosynthesis,

was also over-expressed. On the opposite, mRNA levels for enzymes (TAMM41) involved in cardiolipin (CL) synthesis were not significantly changed. Evidence from the comparison between red- and white-blooded notothenioids is less strong (Table 3). Lipid metabolism of mitochondria requires transfer of phospholipids from the endoplasmic reticulum through specialized contact sites[19]. A family of highly conserved proteins (EMC1-10) appears to be crucial to facilitate such transfer[48]. The majority of transcripts encoding EMC proteins were found to be over-expressed in the icefish muscle transcriptome (Table 3) compared to model species, while the comparison with red-blooded

**Table 2 Muscle differential expression analysis of genes encoding proteins that play a fundamental role in mitochondrial fusion/fission and cristae remodeling.**

| Gene symbols | OG ID | Cmy vs. Red | | Cmy vs. NonAnt | |
|---|---|---|---|---|---|
| | | LogFC | FDR | LogFC | FDR |
| CHCHD3 | OG0000439 | −0.56 | 0.34327 | −1.06 | 0.01315 |
| CHCHD6 | OG0000439 | −0.56 | 0.34327 | −1.06 | 0.01315 |
| IMMT | OG0009745 | −1.01 | 0.03134 | −1.82 | 1.17E-06 |
| DNM1L | OG0000826 | −2.18 | 1.55E-06 | −1.63 | 0.00001 |
| OPA1 | OG0010429 | −0.88 | 0.11476 | −1.47 | 0.00063 |
| MIGA1 | OG0012804 | 1.27 | 0.08039 | 3.08 | 3.6E-12 |
| AK1 | OG0001654 | 2.37 | 0.01166 | 1.75 | 0.00294 |
| AK2 | OG0009063 | 0.56 | 0.21309 | 2.31 | 2.42E-16 |
| AK4 | OG0009643 | 1.83 | 0.00065 | 2.21 | 7.65E-10 |
| AK9 | OG0004716 | 5.10 | 2.96E-06 | 3.33 | 0.0001 |
| CKMT1A mtCK | OG0000821 | −15.43 | 1.68E-24 | −16.04 | 1.41E-27 |

Cmy Chionodraco myersi, Red Red-Blooded Antarctic fish species, NonAnt non-Antarctic species

**Table 3 Muscle differential expression analysis of genes encoding endoplasmic reticulum membrane protein complex (EMC).**

| Gene symbols | OG ID | Cmy vs. Red | | Cmy vs. NonAnt | |
|---|---|---|---|---|---|
| | | LogFC | FDR | LogFC | FDR |
| EMC1 | OG0006378 | −0.25 | 0.65393 | 1.87 | 1.63E-08 |
| EMC2 | OG0010325 | 0.70 | 0.07284 | 1.56 | 4.97E-10 |
| EMC3 | OG0002273 | −1.49 | 0.00033 | −0.64 | 0.05495 |
| EMC7 | OG0008614 | −0.40 | 0.35842 | 0.36 | 0.23869 |
| EMC8 | OG0008125 | 1.98 | 0.00022 | 2.87 | 1.38E-15 |
| EMC9 | OG0008569 | −0.19 | 0.75571 | 0.52 | 0.21265 |
| EMC10 | OG0010959 | 0.08 | 0.89155 | 1.09 | 0.00044 |
| GPAT3/4 | OG0000399 | 0.65 | 0.24223 | 1.46 | 0.00003 |
| GPAT1/2 | OG0007344 | 0.72 | 0.65832 | 0.64 | 0.53293 |
| AGPAT2 | OG0012024 | 1.22 | 0.03150 | 1.31 | 0.00104 |
| MBOAT2/ LPAAT | OG0004179 | 0.65 | 0.21398 | 1.09 | 0.00388 |
| PCYT1A | OG0007558 | 0.56 | 0.23464 | 1.21 | 0.00007 |
| TAMM41 | OG0010111 | −0.35 | 0.63748 | 0.53 | 0.29101 |

Cmy Chionodraco myersi, Red Red-Blooded Antarctic species, NonAnt non-Antarctic species

notothenioids showed no evidence of general upregulation of ECM-encoding transcripts (Table 3).

Several nuclear-encoded genes are potentially involved in the transcriptional control of mitochondrial biogenesis. Tfam (mitochondrial transcription factor A) plays an important role in replication of mtDNA and transcription of mitochondrial genes. In mammals two other regulative proteins, TFB1M, TFB2M (mitochondrial transcription factors B1 and B2) are relevant in mitochondrial genes transcriptional activation[49]. Of these, Tfam and TFB1M appeared to be over-expressed in the muscle of *C. myersi* as compared to the three model species, whilst genes known to activate the mitochondrial machinery like NRF1, NRF2/GABPA, and PGC1α[50], do not appear to be consistently upregulated in the icefish muscle transcriptome (Table 4). It was suggested, however, that the PGC1α - NRF1 pathway may be disrupted in fish due to insertions in the gene sequence of the teleost PGC1α ortholog within the NRF1-binding domain[51]. This is also the case in the *C. myersi* PGC1α that shows a serine-rich insertion in the putative NRF1-binding region. Icefish PGC1α is also mutated at the site responsive to AMPK (AMP-activated protein kinase) which, in mammals, requires a critical threonine

at position 177. AMPK is a major energy sensor implicated in energy balance that in mammals activates metabolic pathways via the PGC1α-NRF1 cascade to respond to energetic challenges when the ATP:AMP ratio is altered[52]. In the icefish, as in other teleosts, threonine is replaced by a proline residue precluding PGC1α phosphorylation by AMPK. Although in fish the AMPK-PGC1α-NRF1 axis seems disrupted, the role of AMPK in responding to muscle metabolic stress was shown to remain central in trout[53]. This could explain the observed remarkable overexpression of AMPK (PRKAA1) in *C. myersi* (Table 4). Similar to AMPK, other master genes of muscle mitochondrial biogenesis, Myocyte enhancer factor 2A (MEF2A), are largely upregulated in the icefish. Also, MEF2A interacts with PGC1α in mammals. MEF2A knock-out mice show disorganized mitochondria with major structural alteration pointing out the central role of MEF2A in mitochondrial biogenesis[54]. Similar evidence was observed for other members of the MEF family (MEF2B, MEF2C, MEF2D; Table 4).

**Comparative analysis of gene duplication**. Massive gene duplication is reported to have affected the evolution of the notothenioid genome[27,29–31]. We compared the genomes of two Antarctic red-blooded (N. coriiceps and D. mawsoni) and two white-blooded notothenioids (C. myersi and C. aceratus). The number of annotated coding genes in the genome of *C. myersi* is 38,140, which is considerably higher than what reported for the closely related icefish species *C. aceratus* (30,773). Likewise, the number of predicted coding genes in *N. coriiceps* (32,331) is significantly higher than what described for *D. mawsoni* (22,516), another red-blooded notothenioid species with a comparable genome size. Fragmentation of genome assemblies is likely the main reason for such discrepancies as a highly fragmented assembly might inflate the estimated number of copies within OGs, because partial gene fragments are counted as two or more copies. Therefore, to control for such phenomenon, we compared all protein-coding genes from all four notothenioid species against the phylogenetically closest species with a high-quality genome, the threespine stickleback, G. aculeatus. Only notothenioid transcripts covering at least 60% of the orthologous sequence in stickleback were retained for duplication analysis (see methods). After such correction, the total number of coding genes was much more similar between species with comparable genome size (C. myersi, 26,805; C. aceratus, 24,900; N. coriiceps, 21,612; D. mawsoni, 19,585). The remaining difference in gene copy number between *C. myersi* and *C. aceratus* was apparently attributable to few gene families (<15), encoding immune-related proteins that

**Table 4 Muscle differential expression analysis of key genes involved in mitochondrial biogenesis.**

| Gene symbols | OG ID | Cmy vs. Red | | Cmy vs. NonAnt | |
|---|---|---|---|---|---|
| | | logFC | FDR | logFC | FDR |
| *TFB1M* | OG0005933 | 2.58 | 0.00007 | 3.58 | 7.52E-18 |
| *TFB2M* | OG0007876 | −0.23 | 0.72366 | 0.11 | 0.84489 |
| *TFAM* | OG0007700 | 0.01 | 0.99426 | 0.77 | 0.02033 |
| *NRF1* | OG0007192 | −1.52 | 0.00821 | −1.36 | 0.00395 |
| *GABPA* | OG0007460 | −0.86 | 0.04306 | −0.33 | 0.36865 |
| *PRKAA1/AMPK* | OG0001835 | 0.50 | 0.26308 | 1.79 | 6.93E-11 |
| *MEF2A* | OG0001443 | 3.26 | 7.36E-08 | 2.43 | 2.65E-11 |
| *MEF2B* | OG0005631 | 1.94 | 0.06032 | 2.37 | 0.00056 |
| *MEF2C* | OG0002939 | 1.94 | 0.00104 | 0.29 | 0.53004 |
| *MEF2D* | OG0003723 | 1.88 | 0.00150 | 1.31 | 0.00076 |
| *PPARGC1A* | OG0004227 | 1.79 | 0.05546 | −0.17 | 0.85610 |

*Cmy* Chionodraco myersi, *Red* Red-Blooded Antarctic fish species, *NonAnt* non-Antarctic species

are reported to have undergone large and rapid lineage-specific expansions in other teleost genomes and are notoriously difficult to correctly annotate. Considering the limited number of gene families accounting for such residual discrepancy, we proceeded by comparing the mean gene copy number for each Orthology Group (OG) between red- and white-blooded notothenioid species. A list of putative duplicated OGs and the corresponding annotations in *C. myersi* is reported in Supplementary Data 1. We then tested whether mitochondrial proteins might be specifically involved in gene duplication events. We observed that genes encoding proteins included in Mitocarta and IMPI were significantly enriched in duplicated OGs between red- and white-blooded species (Fisher-exact test Mitocarta $p = 0.02$, IMPI $p = 0.02$).

Gene duplications in notothenioids have been proposed to increase expression through a gene dosage effect[27,28], although experimental evidence is limited. Here, we compared OGs with $M\Delta \geq 1$, i.e., at least one additional gene copy in white-blooded notothenioids, against OGs with $M\Delta \leq 0$, i.e., genes not duplicated in the icefish genome (*C. myersi* and *C. aceratus*). We observed that duplicated genes were also upregulated (logFC $\geq 1$) more frequently than not duplicated genes (Fisher-exact test $p = 0.01$).

Gene duplication also provides the opportunity for functional specialization of duplicated copies. Tissue-specific expression is considered an indication of such specialization. We thus tested whether genes that underwent duplication specifically in the icefish lineage showed higher tissue specificity compared to not duplicated genes. After calculating the tissue-specificity index $\tau$ (TAU[55]) for all genes across five tissues, genes that belong to those OGs (3601 genes) with a higher number of co-orthologs in *C. myersi* ($M\Delta \geq 1$, i.e., at least one additional copy in the icefish lineage compared to red-blooded notothenioids) showed significantly greater tissue specificity (median $\tau = 0.82$) compared to either genes included in OGs (14,948 genes) in which the number of co-orthologs was equal or lower than other species (median $\tau = 0.77$, Wilkoxon-rank test; $p < 0.00001$) or strictly one-to-one orthologs (1019 genes) (median $\tau = 0.70$, Wilkoxon-rank test; $p < 0.00001$). The distributions of $\tau$-values within the three gene categories are shown in Fig. 3.

**Comparative analysis of promoter regions**. Results of GSEA showed significant enrichment in genes that are putatively regulated by NRF1 and NRF2/GABPA in model vertebrate species. These two transcription factors control the expression of a large set of proteins involved in mitochondrial functioning. Using in silico prediction of putative-binding motifs, we

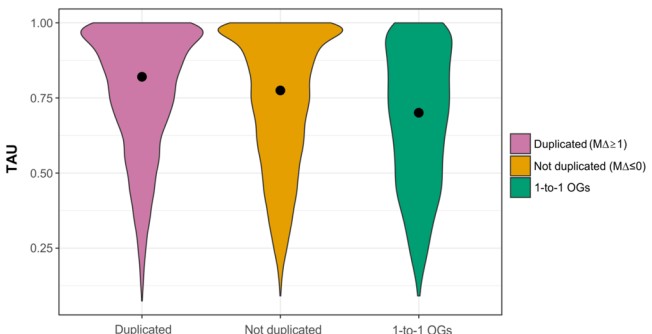

**Fig. 3** Violin plots describing the distribution of $\tau$ (TAU) values within three gene categories.

identified the number of potential binding sites for either NRF1 or NRF2/GABPA in the upstream region of unambiguous 1-to-1 orthologs in zebrafish, tilapia, stickleback, one red-blooded notothenioid (*D. mawsoni*), and one icefish (*C. myersi*). We then compared the number of NRF1 and NRF2/GABPA putative-binding sites in orthologous genes between either *C. myersi* and all three non-Antarctic species or *C. myersi* and *D. mawsoni*. Finally, we tested whether a higher number of putative-binding sites for either NRF1 or NRF2/GABPA in *C. myersi* was associated with significantly higher gene expression (LogFC $> 0$, $p \leq 0.05$) in the icefish muscle. While no significant association was observed for NRF1, for NRF2/GABPA, we found that *C. myersi* genes with at least one additional NRF2/GAPBA-binding site in their promoter compared to model fish species also were more frequently upregulated (Fisher-exact test, $p = 0.003$). The same test between *C. myersi* and *D. mawsoni* was not significant. However, when considering only genes with at least two additional NRF2/GAPBA-binding sites, we observed a significant association with higher expression in the comparison between *C. myersi* and non-Antarctic species (Fisher-exact test $p = 0.01$), as well as between *C. myersi* and *D. mawsoni* (Fisher-exact test $p = 0.03$). Among the genes showing at least two additional predicted NRF2/GABPA motifs in the white-blooded *C. myersi* compared either to non-Antarctic species or to the red-blooded Antarctic species, we found those encoding for WFS1, ZC3H10, and MTG2. The first one of these is a protein involved in mitochondrial-endoplasmic reticulum contacts[56], ZC3H10 is a new mitochondrial regulator[57], and MTG2 is involved in mitoribosome assembly, which is crucial for mitochondrial biogenesis[42].

## Discussion

In this work, we built upon a recent comparison of two genomic regions in the icefish *C. aceratus* that corresponded to the MN and LA globin clusters in other teleost species[31]. Here, we expanded the reconstruction of these two globin gene clusters, including four red-blooded notothenioids and three white-blooded species, defining a new perspective on globin gene loss in the icefish. It has been suggested that the notothenioids only harbor one globin locus—compared to other teleosts containing two—and that this locus was almost completely lost in a single event in the common ancestor of channichthyids, possibly driving the evolution of the hemoglobinless condition[22]. This scenario was partially called into question by Near et al.[23], who however limited their analysis to a single-genomic region. The presence of at least two genomic clusters containing up to ten globin genes in red-blooded notothenioids, reported here for the first time to our knowledge, suggests that the hemoglobin loss was achieved through a series of deletion events and/or rapid sequence divergence due to relaxed purifying selection occurring at separate genomic regions, likely in parallel with progressive reduction of functional importance for oxygen carriers rather than as a single driving mutational event. The irreversible loss of all globin genes is not mirrored by the corresponding deletion of genes involved in the differentiation of the cells that normally carry hemoglobin, the erythrocytes. These genes are downregulated[25], but conserved at the genomic level in white-blooded notothenioids. Likewise, we found no clear evidence for relaxed selection at seven genes encoding proteins with a relevant role in erythropoiesis. A possible explanation is that these genes have a role in other biological processes beside erythropoiesis. It should be noted, however, that orthologous sequences were available only for a very limited number of notothenioid species, which might have reduced the power of the test. As soon as genomes from other red- and white-blooded notothenioids become available, a more powerful analysis of the selective pressures acting on specific icefish genes should be possible.

Loss of hemoglobin is accompanied by unique modifications to favor within-cell oxygen diffusion, through increased mitochondrial density, higher phospholipid content, and altered mitochondrial dynamics[6]. While such adaptations have been investigated at the biochemical and cellular level, the underlying genomic basis has remained elusive so far. Here, we show that a vast transcriptional program significantly involves several pathways controlling key mitochondrial features, with potential effects toward higher mitochondrial biogenesis, increased mitochondrial fusion, modified (reduced) cristae, enhanced transfer between endoplasmic reticulum and mitochondria, and a peculiar lipid profile, with higher phosphatidylcholine and phosphatidic acid (PA), but not cardiolipin (CL). Since PA is considered fusogenic[58] while CL seems to favor mitochondrial fission, the observed expression profile of enzymes involved in lipid biosynthesis is also concordant with marked downregulation of DNM1L/DRP1, which has a key role in mitochondrial fission. The divergent transcriptional profile in the muscle of *Chionodraco* emerging from the comparison with non-Antarctic fish species nicely recapitulates all the cellular adaptions that have been proposed to allow for a life without hemoglobin. The analogous comparison with muscle transcriptome data from red-blooded Antarctic notothenioids provided fully consistent results, with even stronger evidence for a key role of mitochondria in the adaptation to the hemoglobinless condition. In fact, the results obtained from the comparison between *C. myersi* and non-Antarctic species might be explained, at least in part, as the response to the freezing conditions of the Southern Ocean rather than the absence of oxygen carriers. On the opposite, Antarctic red- and white-blooded notothenioids share the same environmental conditions,

therefore, the obtained evidence is the likely consequence of the unique icefish biology. A possible exception is lipid metabolism where divergence at the transcriptome level was less pronounced, which might imply that modifications of lipid metabolism are mostly due to the adaptation to low temperatures. In fact, Chen et al.[30] reported altered lipid metabolism in the Antarctic red-blooded *D. mawsoni* compared to the non-Antarctic notothenioid *E. maclovinus*, although this observation appeared to be mostly linked to increased neutral buoyancy in *D. mawsoni*.

It remains to be assessed whether such articulated transcriptome modifications are the consequence of either genomic divergence or phenotypic plasticity, a long-standing question in evolutionary biology. Looking for evidence of genomic divergence, we tested whether gene duplication might be involved in differential expression of genes encoding mitochondrial proteins. Genes that are duplicated in the *C. myersi* genome compared to other Antarctic red-blooded species showed evidence for enrichment for genetic loci encoding mitochondrial proteins (IMPI or Mitocarta). Gene duplication was significantly associated with higher expression at the mRNA level, as well as to functional specialization in terms of higher tissue specificity. Massive gene duplication has already been shown to characterize the notothenioid genome[27,29–31]. Here, we present compelling evidence that such an evolutionary pattern is particularly important in relation to the peculiar mitochondrial biology of the hemoglobinless icefish and provide a significant link between gene duplication and functional divergence at the whole-transcriptome level.

The transcription factor NRF2/GABPA is known to regulate genes involved in several processes, including mitochondrial biogenesis[59,60]. We found that in the *Chionodraco* genome the number of NRF2/GABPA-binding sites was significantly (Fisher-exact test, $p = 0.003$) higher in the upstream genomic region of genes that are over-expressed in muscle compared to the orthologous region in non-Antarctic, as well as Antarctic red-blooded fish genomes. It has been shown that multiplicity of transcription factor-binding sites in the promoter region generally increases gene expression, with a clear positive relationship between the number of sites and average expression[61,62]. This represents additional strong evidence that the icefish genome has evolved structural modifications that are associated with major changes in the muscle transcriptome profile. Therefore, genomic divergence seems at least partially responsible for the unique cellular adaptations that allow the hemoglobinless icefish to survive. While working on Antarctic fish biology remains difficult for obvious reasons, the popularization of genomic tools, such as comparative transcriptome analyses and functional genomic assays, holds the promise to help uncover the molecular mechanisms underlying the evolution of this unique group of vertebrates.

## Methods

**Ethics**. No specific permits were required for the work described here. Animals included in the present study were not subjected to any experimental manipulation. The study was performed in accordance with the EU directive 2010/63/EU and Italian DL 2014/26. Experiments and killing procedures were monitored and carried out by authorized staff to minimize animals' suffering.

**Ortholog analysis**. Homology relationships between *C. myersi*, and model teleost genomes were reconstructed with the software OrthoFinder[63] using a total of 19 annotated genomes. OrthoFinder uses the BLAST tool[64] to compute sequence similarity scores between sequences in multiple species and then uses the MCL clustering algorithm[65] to identify groups of highly similar sequences within this dataset. Protein datasets from Ensembl or NCBI databases were downloaded for *Astyanax mexicanus* (Ensembl GCA_000372685.2), *Danio rerio* (Ensembl GRCz11 GCA_000002035.4), *Gadus morhua* (Ensembl gadMor1), *Gasterosteus aculeatus* (Ensembl BROAD S1), *Oreochromis niloticus* (Ensembl GCA_000188235.1), *Oryzias latipes* (Ensembl GCA_002234715.1), *Poecilia formosa* (Ensembl Poecilia_formosa-5.1.2), *Takifugu rubripes* (Ensembl GCA_000180615.2), *Tetraodon nigroviridis*

(Ensembl TETRAODON 8.0), *Xiphophorus maculatus* (Ensembl GCA_002775205.2), *Lepisosteus oculatus* (Ensembl GCA_000242695.1), and *Larimichthys crocea* (NCBI Release 102). The proteomes of *Sparus aurata*[66] along with the closely related European seabass (*Dicentrarchus labrax*, downloaded from http://seabass.mpipz.mpg.de/cgi-bin/hgGateway), five Antarctic fish *Parachaenichthys charcoti* (Bathydraconidae)[35], *Notothenia coriiceps* (Nototheniidae)[37], *Dissostichus mawsoni* (Nototheniidae)[30], *Eleginops maclovinus* (Eleginopsidae)[30], and *Chaenocephalus aceratus* (Channichthyidae)[31] were added in the analysis. The orthology groups (OGs) were calculated for all species using OrthoFinder default parameters.

In order to reduce the effect of the fragmentation of Antarctic genome assemblies on the estimated number of duplicated genes, all protein-coding genes of *C. myersi*, *D. mawsoni*, *N. coriiceps*, *C. aceratus*, and *E. maclovinus* were compared to the proteome of *G. aculeatus* (Ensembl BROAD S1) by BLASTP analysis. Only proteins covering at least 60% of the orthologous sequence in stickleback were employed in the final datasets (Supplementary Table 7).

After a first round with OrthoFinder and all the 20 species, the *P. charcoti* dataset was excluded due to the low-quality of sequence data resulting in unreliable protein alignments.

Protein-coding gene gain and loss between icefish and red-blooded Antarctic species were then assessed by employing Mean Δ ($M\Delta$) metrics: mean number of *C. myersi* and *C. aceratus* genes—mean number of *N. coriiceps* and *D. mawsoni* genes.

**Comparative analysis of muscle transcriptomes.** In order to compare muscle gene expression across different species, RNA sequencing (RNA-seq) datasets were retrieved from the NCBI Sequence Read Archive. RNA-seq libraries of muscle from *D. rerio*, *O. niloticus*, and *G. aculeatus* were downloaded (Supplementary Table 8). For each species, five biological replicates were retrieved. RNA-seq data from three Antarctic species (*D. mawsoni*, *N. coriiceps*, *P. charcoti*) were also used for RNA seq analysis, for a total of four replicates representing the red-blooded Antarctic sample group (Supplementary Table 8).

The reads of each species were mapped against the corresponding assembled genome by means of STAR aligner[67] and following the *two-pass* mapping mode. The maximum number of mismatches was fixed to 6% and a threshold for multiple matches was set to 200 (–outFilterMultimapNmax 200). Read counts for each sample, at the gene level, were extracted by GeneCounts quantification while running STAR.

Analysis of differential gene expression was conducted in EdgeR[68]. For each sample, the sum of raw cpm (count per million) of all genes belonging to the same OG was calculated by means of *aggregate* function in R v. 3.5.3. OGs showing a cpm value <1 in more than half of the samples for each species were filtered out. Remaining OGs ($n = 9721$) were normalized with the Trimmed Mean of *M*-values (TMM) method and, after estimating common and tagwise dispersions, likelihood-ratio test (lrt) as implemented in EdgeR was employed to assess differentially expressed genes (DEGs) between *C. myersi* and the other three model species, as well as between *C. myersi* and the red-blooded Antarctic samples.

OGs with $\log_2$ fold-change (logFC) $< -1$ or $>1$ and false-discovery rate (FDR) $\leq 0.05$ were considered significant.

**Gene-set enrichment analysis.** GSEA[69] was used to identify enriched functional categories within OGs differentially expressed in *C. myersi* compared to other teleost species.

The GSEAPreranked v1[69] tool was applied to RNA-seq data used in the comparative transcriptomic analysis (9721 OGs). Ranking values were assigned to OGs based on EdgeR lrt *p*-value as follows: (i) logFC > 0, score = 1−*p*-val; (ii) logFC < 0, score = −(1-*p*-val). The enrichment analysis was carried out by setting Enrichment statistic = Classic, Normalization mode = meandiv, and Number of permutations = 1000. A panel of 25 Gene sets was interrogated comprising BIOCARTA, KEGG, and GO sets related to mitochondrial biogenesis and functioning as downloaded from the Molecular Signatures Database v6.1 (MSigDB), http://software.broadinstitute.org/gsea/msigdb), as well as two gene sets, IMPI (Integrated Mitochondrial Protein Index) and MitoCarta 2.0[70] genes, which were downloaded from MitoMiner 4.0 database[71].

**Assessment of duplication enrichment on gene sets.** All OGs were used in order to investigate whether the above mentioned gene sets were significantly enriched in OGs that duplicated in the icefish compared to red-blooded notothenioids. To this end, the $M\Delta$ metric was used. All OGs with $M\Delta \geq 1$ were considered duplicated. All genes that composed a single OG were retrieved and, whenever possible, multiple Gene Symbols were retained. In case of multiple Gene Symbols for the same OG, they were assigned the same $M\Delta$ value. A Fisher's exact test was then used to assess whether gene-set showed a significant difference in the frequency of duplicated OGs between icefish and red-blooded notothenioids.

**Expression breadth.** Tissue specificity (i.e., expression breadth) of OGs was estimated using the $\tau$ index[55]. The index $\tau$ ranges from 0 to 1 and it is defined as $\tau = \sum N_i = 1(1 - x_i)/N - 1$, where $N$ is the number of tissues and $x_i$ is the expression profile component normalized by the maximal component value

($\tau = 1$ single tissue expression, $\tau = 0$ ubiquitous expression). Tissue specificity was calculated on the normalized expression (TMM normalized log2 cpm) evaluated in brain, liver, skeletal muscle, kidney, and spleen from RNA-seq data produced in the present study (see Supplementary Table 8). Median values of $\tau$ were calculated on OGs found duplicated ($M\Delta \geq 1$) in the icefish genome compared to not duplicated OGs ($M\Delta \leq 0$) and 1-to-1 OGs. In order to assess the significance of inter-group $\tau$ differences, a pairwise Wilcoxon rank-sum test was implemented in R.

**Transcription factor-binding sites distribution.** Position frequency matrices (PFM) of transcription factor (TF)-binding profiles for NRF2/GABPA were downloaded from the JASPAR database (http://jaspar.genereg.net/). The presence and number of TF-binding sites (TFBS) were investigated in 1-to-1 orthologues (n = 3,895) between *C. myersi*, *D. mawsoni*, and the three model species *D. rerio*, *O. niloticus*, and *G. aculeatus*. A region of length 5 kb upstream the putative transcription starting site of these orthologous genes was then used as input for FIMO[72], a MEME[73] suite's tool (Motif-based sequence analysis tools). For each gene, $M\Delta$ was calculated between the number of TFBS found in *C. myersi* and the median TFBS number across non-Antarctic species. The same comparison was carried out between *C. myersi* and *D. mawsoni* (number of *C. myersi* TFBS—number of *D. mawsoni* TFBS). A Fisher's exact test was performed in order to test for significant differences in the $M\Delta$ distribution between over-expressed 1-to-1 orthologous genes and not differentially expressed in *C. myersi*.

**Annotation of hemoglobin clusters in Antarctic fish genomes.** The LA and MN hemoglobin clusters were identified in the genome of *C. myersi*, *C. hamatus*, *N. coriiceps*, *P. charcoti*, *C. aceratus*, *D. mawsoni*, and *E. maclovinus* based on the genomic organization described by Opazo et al.[24]. There is a high-degree conservation between five genes to the left of the MN cluster: *aanat*, *mgrn1*, *rhbdf1a*, *mpg*, and *nprl3*. The two genes to the right, *kank2* and *dock6*, are also well conserved. The LA cluster is less conserved with the gene *rhbdf1* to the left, *aqp8* and *lcmt1* on the right side[24]. For both clusters, *N. coriiceps* protein sequences of conserved flanking genes, together with α- and β-globin genes, were downloaded from the NCBI database and employed as query in BLASTp searches ($10^{-5}$ e-value) against the draft genomes of the six species. Contigs containing target genes were then further employed for manual annotation, using BLASTp output in the software UGENE[74], to draw the putative LA and MN clusters.

One-to-one orthologs for seven genes coding for key erythropoiesis factors from six teleost species (*D. rerio*, *G. aculeatus*, *O. niloticus*, *E. maclovinus*, *D. mawsoni*, *C. myersi*) were aligned using a codon-guided alignment software (TranslatorX)[75] available at http://translatorx.co.uk/. The algorithm MAFTT with default settings was selected for alignment in TranslatorX and codon-based alignment were refined using GBlocks. Aligned coding sequences were then processed with the software RELAX[39], available at https://www.datamonkey.org/relax. In RELAX, the tree branch leading the icefish species (*C. myersi*) was compared to the remaining branches to test whether relaxed selection was detected as a consequence of hemoglobin loss.

**Statistics and reproducibility.** Protein-coding gene gain and loss analysis was assessed by employing Mean Δ ($M\Delta$) metrics: mean number of icefish genes – mean number of red-blooded Antarctic species genes.

Gene expression analysis was conducted using five biological replicates for non-Antarctic species and icefish while four replicates were employed for red-blooded Antarctic species. Analysis of differential gene expression was conducted in EdgeR. For each sample, the sum of raw cpm (count per million) of all genes belonging to the same OG was calculated by means of aggregate function on R. OGs showing a cpm value <1 in more than half of the samples for each species were filtered out. A likelihood-ratio test (lrt) as implemented in EdgeR was employed to assess differentially expressed genes considering significant a $\log_2$ fold-change (logFC) $< -1$ or $>1$ and false-discovery rate (FDR) $\leq 0.05$.

In order to assess duplication enrichment on specific gene sets a Fisher's exact test (FET) was conducted considering as duplicated all OGs having $M\Delta$ values > 0. Gene sets with FET *p*-value $\leq 0.05$ were considered significantly enriched.

FET was also employed in order to assess if duplicated OGs were more frequently over-expressed than OGs not duplicated. In this case, OGs with $M\Delta \geq 1$ were compared to OGs with $M\Delta \leq 0$ and overexpression cutoff was set to logFC $\geq 1$.

**Reporting summary.** Further information on research design is available in the Nature Research Reporting Summary linked to this article.

## Data availability

*Chionodraco myersi* and *Chionodraco hamatus* genome assemblies were deposited in DDBJ/ENA/GenBank under the accession RQJG00000000 (version RQJG01000000) and RRCA00000000 (version RRCA01000000), respectively. All raw sequence data produced in this study were deposited in NCBI Short Reads Archive (SRA) under accession numbers from SRR8197047 to SRR8197058. Details are reported in Supplementary Table 8.

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

## Acknowledgements

This paper is dedicated to the memory of our colleague Dr. Guido di Prisco, a pioneer in Antarctic science, who recently passed away. This research was supported by Italian PNRA 2013/A21.10, PNRA 16_00226, PNRA 16_00099, and PNRA 2016_00307. We would like to thank Nils Koschnick (AWI) and Emilio Riginella (Stazione Zoologica Anton Dohrn, Napoli) for collecting the samples during "Polarstern" ANT-PS82 and ANT-PS86 cruises. We are indebted to the Polarstern crew for their indispensable help during fishery activities. C.P. acknowledges financial support from the University of Padua (BIRD164793/16) and from the European Marie Curie project "Polarexpress" Grant No. 622320.

## Author contributions

T.P. and L.B. conceived the study. M.L., F.C.M. and C.P. provided the fish. R.C. isolated the *C. myersi* and *C. hamatus* genomic DNA and prepared genomic and RNA-seq libraries. M.B. and N.V. performed the genome assembly and the annotation of the two *Chionodraco* genomes. M.B. and S.F. performed the homology and gene family expansions/contractions analyses. M.B. and S.F. performed the molecular evolution and comparative transcriptomic analyses of icefish data. L.B. oversaw molecular evolution data analysis. C.P., G.S., M.P., M.L., F.C.M., N.V. and L.Z. participated in data interpretation and in the manuscript writing. M.B., S.F., L.B. and T.P drafted the paper. All authors read, amended, and approved the final manuscript.

## Competing interests

The authors declare no competing interests.
