## [Peer Review File · Communications Biology]

Reviewers' comments:

Reviewer #1 (Remarks to the Author):

The paper describes whole genome sequencing of two Antarctic icefish, *C. myersi* and *C. hamatus*, and made comparisons of *C. myersi* muscle transcriptome to three teleost fishes. While the paper is of some value in terms of understanding of genome evolution in the Antarctic marine environment, the paper need to be significantly improved before it can be accepted for publication.

1. Issues on the genome assemblies:

1). The quality of *C. hamatus* genome assemble does NOT qualify for publication. The assembled *C. hamatus* genome size of 0.83 Gb accounted merely half of the estimated size of an icefish. More unsatisfied is the short contig N50 of this assemble, which is only 2.7 kb in length. No significant conclusions can be drawn from such a low quality genome. As admitted by the authors, they are not using this assemble for in-depth comparisons in this manuscript. I recommend the authors to either significantly improve the assemble or remove it from this manuscript to strengthen the data.

2). The authors reported a size of the *Chionodraco myersi* genome to be 1.12 Gb, which is significantly smaller than reported genome sizes of the species in the Channichthyidae family (Genome Enablement of the Notothenioidei: Genome Size Estimates from 11 Species and BAC Libraries from 2 Representative Taxa. *J Exp Zool B Mol Dev Evol.* 314(5): 369–381), Which ranged from 1.6 to 1.8 Gb. The authors should explain the difference and estimate the genome size correctly. In our knowledge, distribution of k-mer occurrence cannot give a correct estimation if a high heterozygosity rate is present in the genome, as indicated by the bimodal k-mer curve the authors showed in the manuscript.

3). The current *C. myersi* genome assembly still left room for improvement. A scaffold N50 length under 50Kb is relatively short when the Pacbio long read sequencing technology has been adopted in genome sequencing. The authors stated they used a hybrid strategy for assemble, but only the MaSuRCA genome assembler is mentioned in the manuscript. The hybrid assemble strategy should be described in more detail. I wonder if other assemble algorithms have been implemented in current genome assemblies.

2 issues on the transcriptome comparison:

The authors obtained the *C. myersi* skeletal muscle transcriptome and compared to muscle transcriptomes of zebrafish, Tilapia and stickleback in the attempts to address the genomic basis for the icefish-specific increase of mitochondrial density and other physiological adaptations. However this comparison is far from convincing due to lack of samples from red-blooded Antarctic notothenioids and due to the limited tissue types (only one tissue) incorporated in the comparison. For example, elevation of GPATs and lipid synthesis in the icefish muscle transcriptome observed by the authors might merely resultant of cold environment, rather icefish specific adaptation to their hemoglobinless condition. Alterations in lipid synthesis is a reported adaptation in other red-blooded notothenioid species (The genomic basis for colonizing the freezing Southern Ocean revealed by Antarctic toothfish and Patagonia robalo genomes. *Gigascience.* doi: 10.1093/gigascience/giz016, 2019). Authors also linked increased NOS1 transcripts to possible increase of NO levels in the icefish. This again lacks solid evidence when only one tissue is surveyed and the protein level of the enzyme is not assayed. I suggest the authors include the transcriptomes of other red-blooded Antarctic notothenioids and expand the comparison to other tissues.

3. Issues in gene duplication:

The authors compared the *C. myersi* Orthologous groups to other teleosts and drew conclusion that gene duplication might have contributed to mitochondrial biogenesis. Recently, three more notothenioid genomes including one icefish have been published (Antarctic blackfin icefish genome reveals adaptations to extreme environments. *Nat Ecol Evol.* 3(3):469-478, 2019; The genomic basis for colonizing the freezing Southern Ocean revealed by Antarctic toothfish and Patagonia robalo genomes. *Gigascience.* doi: 10.1093/gigascience/giz016, 2019). The authors should include these species in gene duplication analysis. Including of these species is necessary to address the genomic basis of icefish-specific phenotypes as the authors attempted throughout this manuscript.

Reviewer #2 (Remarks to the Author):

Dear authors,

After reading through your manuscript (COMMSBIO-19-0148-T) entitled "Comparative analysis of the icefish genome and transcriptome reveals the key role of mitochondria for a life without hemoglobin at subzero temperatures" my recommendation to the Editor is to accept with major revision.

First, I want to highlight that I think this is a nice piece of work that would be a nice addition to the already published literature on the icefish genetics/genomics in relation to adaptation to freezing temperatures (e.g. Bo-Mi et al. *Nat Ecol and Evol* 2019, Baalsrud et al. *MBE* 2018). In this regard, I would strongly urge the authors to take a closer look at this recently published paper in *Nat Ecol and Evol*, and refer to this work where appropriate.

The paper is well written and the analyses conducted are adequate for the questions in mind. However, I have some questions regarding the presentations of the analyses conducted as well as some suggestive additional analyses they should look into before the authors resubmit the ms, which I think will improve the paper further.

First, I have no concern regarding the genome assemblies. Think, that from the sequence data generated – they have produced OK draft genomes that could be used for detection of gene losses and gene expansions. However, it could have been nice with a table in the Supp – where some comparisons with other published genomes were highlighted. The *C. myersi* genome assembly is not that bad compared to many other teleost genomes out there.

But that said, the number of identified genes is in the upper segment (i.e. 38,127 putative protein-coding genes), and thus, for some of the analyses conducted within this ms caution should be made. Especially in regard to the gene duplication analysis.

Earlier reports have shown that differences in genome assembly qualities would impact the estimation of copy number of large gene families (e.g. Tørresen et al. *BMC Genomics* 2017).

In light of the high number of duplicated genes, it would be nice with some more validations of that number. For instance, you could set up some criteria for the genes that are included in the final annotation:

- 1) Genes predicted only by ab-initio programs: these genes were considered good only if confirmed by at least four different ab-initio programs, if they were complete (with a start and a stop codon) and longer more than 300 base pairs.
- 2) Gene supported only by external evidence (e.g. proteins/RNA-seq): they needs to be confirmed by at least two different evidence or by one external evidence and at least three different ab-initio gene predictors.

3) Predicted genes with a low ab-initio support (filter described at step 1) were further processed. Genes supported by less than four ab-initio programs were search against a database of teleost protein sequences. Proteins with a sequence coverage match higher than 70% and an e-value lower than $1E-20$ were recovered.”

None of these criteria guarantee that the genes are full-length, but criteria 1 and 2 should give long gene models. Could you validate the number of gene duplicates in some way? For instance, reporting the number of genes that have more than 70 % (80 % and 90 % maybe also) sequence coverage match against that database of teleost protein sequences. There might be other ways too, but the important factor is to avoid inflating the number by reporting parts of genes (which could happen, even with your criteria). You could have situations where you report the first 45 % of a gene as one gene, and the next 55 % as another.

Second, for the reconstruction of the LA and MN clusters in the different species (included some low coverage genomes already published) I see the need for a more detailed description both in the results section as well as in the methods. Especially since – as the authors state – some of the genomes used are highly fragmented. How the authors extracted the contigs from the different genome assemblies (how many unitigs/contigs in total was needed to do the reconstructions for the different species, and would be good to see the making of the clusters in the drawings where you f. ex. show the overlapping contigs and where they do not overlap, where you have the gaps). Further, an additional Supp table with the statistics re the reconstructions for the different genome assemblies linked to their publication (in addition to the added info in Fig 1 and in the text (as described above)) would be a nice add on as well.

As is, this information is totally lacking (from the methods/results and Supp).

Additionally, the fact that the authors used the genomic organization of the LA and MN clusters as given by Opazo et al 2013, if would have been great to see them discuss if this could give rise to conflicting results or if they were able to identify the gene order within the two different gene clusters for the different spices with high confidence (based on the contigs/unitigs extracted from the genome assemblies).

These additions to the paper, would be highly valuable for the reliability/reproducibility of the methods/analyses conducted as well as a good way of showing others how to make use of these low coverage genomes.

And one last comment regarding this issue: Maybe the authors here should take a closer look at the recently published blackfin icefish genome in Nat Ecol and Evol. (also mentioned above to further confirm that their method is reliable).

Third, I want to point out minor issues regarding the transcriptome analyses conducted. I have no comments on the analyses themselves, but I would have appreciated some more details around the sequencing data downloaded for the three species (*D. rerio*, *O. niloticus* and *G. aculeatus*). As is, the information given in the Supp table is scarce – here I would like to see information about the sequencing technology used (e.g HiSeq500 vs HiSeq4000 or maybe even HiSeq2000?), number of reads, number of replicates and so on for the different species.

Why? The main reason for this, is that it is shown that you easily detect differences between runs/library preps (and especially between different technologies) – that are then due to the runs/technologies and not differences due to biology. Even if I trust the differences found/reported here are due to difference in biology – I think this is something the authors could bring up in the discussion.

Further, I find it highly noteworthy that they did the differentially expression analyses both as a comparison between icefish and all non-Antarctic species (based on the separation on the first component (PC1) as well as only using one of them (*G. aculeatus*). The similarity in results clearly demonstrate that pooling of the three non-Antarctic species is scientifically valid (since you actually

see that there are some differences between these species given on the second component (PC2). So, for this, I would encourage the authors to summarize these results in the Supp as a nice validation of the comparison given in the main text (as I can see this information is not included in the current version of the ms or the Supp).

Then to some of the results that could be looked further into. F. ex. the authors state in line 140-141: "Unlike the fate of the two globin clusters, inspection of the *C. myersi* genome revealed that a full-length copy is retained for all genes encoding key erythropoietic transcription factors that were found to be suppressed (full list in Table S2 of Xu et al. 201525) in the icefish (data not shown)".

Could it be that the downregulation of these genes (due to living in freezing environments and no need for erythropoietic cells) further gave rise to the relaxed purifying selection on the Hb regions (as the authors here indicate with sequence decay). In other words what happened first? Maybe gene expression data from the red-blooded notothenioids, would be of added value?

Similar, the result discussed in line 316-325:

"The transcription factor NRF2/GABPA is known to regulate genes involved in several processes including mitochondrial biogenesis^{61,62}. We found that in the *Chionodraco* genome the number of NRF2/GABPA binding sites was significantly higher in the upstream genomic region of genes that are overexpressed in muscle compared to the orthologous region in non-Antarctic fish genomes. It has been shown that multiplicity of transcription factor binding sites in the promoter region generally increases gene expression, with a clear positive relationship between the number of sites and average expression^{63,64}. This represents a second, much stronger evidence that the icefish genome has evolved structural modifications that are associated with major changes in muscle transcriptome profile. Therefore, genomic divergence seems at least partially responsible for the unique cellular adaptations that allow the hemoglobinless icefish to survive".

Here, number of binding sites for these above-mentioned genes could be easily looked into for the red-blooded notothenioids (that they do have data for), which could give added value on the speculations if these genetic signatures are specific for the icefishes or not.

Lastly, I would also recommend the authors to modify the title (since they seemingly here refer to one icefish genome? – but have in fact generated two – and used additional genome information from already published genomes). The title is:

"Comparative analysis of the icefish genome and transcriptome reveals the key role of mitochondria for a life without hemoglobin at subzero temperatures"

Would suggest to change to:

"Draft icefish genome assemblies and transcriptome data reveal key role of mitochondria for a life without hemoglobin at subzero temperatures"

Some additional more detailed comments that should be taken under considerations:

Line 90: "Through the analysis of the first draft genome of two white-blooded channichthyid species,.....

Should be re-written to:

"Through comparative genomic analyses of draft genome assemblies of two white-blooded

channichthyid species,.....

See that you throughout the paper often refer to "comparative analysis" – I guess you did multiple analysis – and should do this quote in plural: "comparative analyses". And preferably also add genomic, i.e. comparative genomic analyses.

Line 283-285:

".....It was previously supposed that a single globin locus was present in notothenioids which was almost completely deleted in a single event in the common ancestor of channichthyids, possibly driving the evolution of the hemoglobinless condition²²"

Think this sentence is a bit vague – would suggest to re-write:

"It has been suggested that the notothenioids only harbor one globin locus – compared to other teleosts containing two – and that this locus was almost completely lost in a single event in the common ancestor of channichthyids, possibly driving the evolution of the hemoglobinless condition"

Then for line 287-289: It should be made clear that you here refer to your own results/data.

Line 415-417: This sentence gives very little information – more details should be given here and in the result section. And the sentence as such is poorly written:

"LA and MN hemoglobin clusters were searched in *C. myersi*, *C. hamatus*, *N. coriiceps*, *P. charcoti*, *C. aceratus* and *E. maclovinus* genomes following the genomic organization described by Opazo and colleagues²⁴".

Here you should describe in more detail how you did your BLAST search (e-value threshold) and also in more detail which flanking genes (the genomic organization given by Opazo et al).

Reviewer #3 (Remarks to the Author):

This study entitled "Comparative analysis of the icefish genome and transcriptome reveals the key role of mitochondria for a life without hemoglobin at subzero temperatures" presents new genome assemblies for two Antarctic icefish species as well as gene expression data for one of the two species to investigate the evolutionary loss of hemoglobin in these species. As this loss is a unique feature of icefishes among all vertebrates and furthermore one that may appear detrimental to the species but nevertheless is found in a rapidly radiating clade, the insights into its emergence are of great interest to evolutionary biologists. In particular, the results of this study show that the loss did not occur as a single event, but proceeded in multiple steps. This work thus lays an important foundation for future studies, that could employ genomic data of further icefish species to elucidate the details of this step-wise loss.

While I am not an expert on some of the applied workflows (e.g. the gene-expression comparisons), all of them appear to be done in an expertly manner (and some were established in a recent paper in Nature Communications Biology). The manuscript is well-written, but some details could be clarified as I will specify below. Besides these details, there is only one part of the study that I find somewhat

disappointing, namely the contiguity of the genome assemblies. The genome assemblies of *Chionodraco myersi* and *C. hamatus* are based on extremely high coverage of 80-100x 150 bp Illumina reads (plus PacBio for *C. myersi*) and the genome sizes of the two species are not extreme (0.8-1.1 Gb), but nevertheless, the assemblies are quite fragmented. The one for *C. hamatus*, based on 80x Illumina data, has a N50 contig length of 2700 bp. In comparison, Malmstrøm et al. (<https://www.nature.com/articles/sdata2016132/tables/3>) sequenced the genome of the very closely related icefish species *Chaenocephalus aceratus* with a coverage of 36x 150 bp Illumina reads and obtained a contig N50 of 5500 bp. Moreover, the *C. hamatus* assembly presented in this study has nearly 40% missing data in scaffolds. I wonder if the assembly strategy with the CLC Genomics Workbench is not ideal and better assemblies could be obtained with other tools. I see no reason why the 80x data for *C. hamatus* should not yield a contiguity that is at least as good as that for *Chaenocephalus aceratus* obtained by Malmstrøm et al. with 36x data. As the genome assemblies presented in this study will provide an important resource for the community, I urge the authors to also try other assembly strategies in order to optimize the assembly quality. Even though the assembly of *C. myersi* is not as fragmented as the one for *C. hamatus*, I would also expect that still better contiguity should be possible with the 100x Illumina read data plus PacBio reads, so I also suggest that the authors attempt to improve this assembly. At the very least, the choice of assembly tools should be justified in the manuscript, and particular reasons for the low contiguity should be discussed. That said, I note that the current low assembly contiguity apparently did not limit the permitted conclusions.

I also feel that the manuscript is currently a bit poor in figures. Better illustrations will attract more readers, thus, I suggest that at least a third figure is added. If images of the two species are available, these could be shown, other options include a map with the geographic ranges of the species, a more detailed image of the relationships and divergence times of the species, or some plots of the data that is currently presented only in tables in the main text and the supplement. All of these are just suggestions.

Overall, I find this study valuable and interesting and think that it may become suitable for publication in Nature Communications Biology. A list of minor comments, in order of their appearance, is provided below.

Minor comments:

- l. 18; "vertebrate": I would use plural to clarify that there are multiple icefish species.
- l. 31: This sentence is slightly confusing because it sounds as if the family was first described by Ruud, but Ruud was only the first to describe their lack of hemoglobin.
- l. 40; "is not sufficient": It should be clarified for what this is not sufficient.
- Results section "Icefish genome assembly and annotation": Might be worth repeating here what sequencing technology was used.
- l. 110: I suggest mentioning here which other notothenioid genomes were used.
- l. 114; "notohenioids": Typo.
- l. 116; "Bathidraconidae": Typo.
- l. 120; "carried on": This should be "carried out on".
- Figure 1: A reference should be given for the phylogeny. The meaning of the asterisk, being the region displayed in B, should be mentioned. The meaning of the arrows in B should be explained in the figure legend, not just the main text.
- l. 161-163: These expression differences do not necessarily have anything to do with hemoglobin loss. They could be related to many other differences between icefishes and the other taxa, including cold adaptation. I think this should be pointed out here or elsewhere.

- l. 164, "FDR", "NES": Abbreviations should be explained.
- l. 170: It should be explained how these 16 genes were identified as being involved in mitoribosome assembly (presumably from the literature/databases). This also applies to the genes listed in Tables 2-4. It should be clarified if these lists are exhaustive or a selection.
- l. 175: Period after "myersi" should be removed.
- l. 178; "significant": The adverb should be used.
- l. 182: A reference is needed for the comment on the controversial role.
- l. 205; "among which": Word missing.
- l. 208: I don't think "justified" is the right word here.
- l. 210: I wouldn't say that this explains those phenotypes, but it is consistent with (or linked to) them.
- l. 217; "resulted overexpressed": Grammar not correct. "resulted" should be replaced, e.g. with "appeared".
- l. 220: The word "the" is missing before "that PGC1 α ".
- l. 228; "was showed": Should be "was shown".
- l. 236; "generates": Should be "generate".
- l. 240: Reference missing for the statement.
- l. 247; "the first one": "of these" could be added.
- l. 333: Accession numbers should be provided for the 18 genomes, as for some of these species multiple assemblies exist on NCBI and ENSEMBL.
- l. 335; "Protein dataset": Should be plural.
- l. 341: Notothenia is not a member of Channichthyidae.
- l. 351; "Both metrics...": This is of course not surprising given that the one is simply a fraction of the other and the distinction is only between positive and negative values. So, this should maybe be rephrased. And WMDelta is not provided anywhere it seems, but it could be added to Table S7.
- l. 372; "lrt_CmyvsAll": This label is not reused, so it could be removed.
- l. 395, 396; "expression breath": Typo, should be "expression breadth".
- l. 409; "arbitrary region": I'd rephrase. Not the region was arbitrary, just it's length.
- l. 413: "C. myersi overexpressed": Grammar unclear, I'd rephrase as "OGs overexpressed in C. myersi".
- l. 418: "blast" should be written in upper case.
- l. 472: Names missing in reference.
- Table 1: Abbreviations LSU/SSU (presumably large/small subunit) should be explained.
- Supplementary Material, throughout: Version numbers should be given for all programs.
- Supplementary Material, page 3; "a fish protein database": Details about this database should be provided, also for databases mentioned on page 5.
- Supplementary Material, all tables: Short table captions would be helpful.
- Supplementary Material, section "Gene prediction": There are lots of typos and grammar mistakes in this section. It should be proofread carefully.
- Supplementary Material, Table S3: Rows with numbers of scaffolds/contigs > 10M nt can be deleted. And I don't understand how the shortest contig can have a length of 0.
- Supplementary Material, Table S7: Information missing: What are the numbers in this table. Also abbreviations "GS", "OG_ID", "DeltaM" should be explained. And instead of "DeltaM" it should probably be "MDelta".
- Supplementary Figure S1: Not sure where I am supposed to see the kmer-peaks indicated with dashed lines.

Authors reply to reviewers' comments:

AU: *We thank all three reviewers for their constructive comments, which helped us to prepare what hopefully is an improved version of our manuscript. As a general comment, we would like to point out that the previous version of this manuscript was submitted before the publication of two relevant papers (Kim et al. 2019, Chen et al. 2019), which reported the draft genome of three notothenioid species, including one icefish. Therefore, we could not include such novel information in the submitted manuscript. However, as recommended by reviewers 1 and 2, we have now entirely revised our analyses incorporating the results from Kim et al. 2019 and Chen et al. 2019.*

Kim, B. M. et al. Antarctic blackfin icefish genome reveals adaptations to extreme environments. *Nature Ecology & Evolution* 3, 469 (2019).

Chen, L. et al. The genomic basis for colonizing the freezing Southern Ocean revealed by Antarctic toothfish and Patagonian robalo genomes. *Gigascience* 8, (2019).

Reviewer #1 (Remarks to the Author):

The paper describes whole genome sequencing of two Antarctic icefish, *C. myersi* and *C. hamatus*, and made comparisons of *C. myersi* muscle transcriptome to three teleost fishes. While the paper is of some value in terms of understanding of genome evolution in the Antarctic marine environment, the paper need to be significantly improved before it can be accepted for publication.

1. Issues on the genome assemblies:

1). The quality of *C. hamatus* genome assemble does NOT qualify for publication. The assembled *C. hamatus* genome size of 0.83 Gb accounted merely half of the estimated size of an icefish. More unsatisfied is the short contig N50 of this assemble, which is only 2.7 kb in length. No significant conclusions can be drawn from such a low quality genome. As admitted by the authors, they are not using this assemble for in-depth comparisons in this manuscript. I recommend the authors to either significantly improve the assemble or remove it from this manuscript to strengthen the data.

AU: *We agree. Every mention to the C. hamatus genome draft has been removed from the main text. However, we still keep a brief description of the sequencing and assembly of this genome in the Supplementary Information as we believe that despite the limited quality, making such information publicly available might still be of useful for scientists working on Antarctic fish.*

2). The authors reported a size of the *Chionodraco myersi* genome to be 1.12 Gb, which is significantly smaller than reported genome sizes of the species in the Channichthyidae family (Genome Enablement of the Notothenioidei: Genome Size Estimates from 11 Species and BAC Libraries from 2 Representative Taxa. *J Exp Zool B Mol Dev Evol.* 314(5): 369–381), Which ranged from 1.6 to 1.8 Gb. The authors should explain the difference and estimate the genome size correctly. In our knowledge, distribution of k-mer occurrence cannot give a correct estimation if a high heterozygosity rate is present in the genome, as indicated by the bimodal k-mer curve the authors showed in the manuscript.

AU: *We agree that the estimated genome size for C. myersi is lower than the estimate obtained by Detrich et al. (2010). However, similarly lower are the size estimates reported by Kim et al. (2019) for another icefish*

species (1.1 Gbp) and for two red-blooded notothenioids (0.73-0.84 Gbp, Chen et al. 2019). All these estimates are at variance with what reported by Detrich et al. (2010). We now briefly describe and discuss such discrepancy in the text (results section)

3). The current *C. myersi* genome assembly still left room for improvement. A scaffold N50 length under 50Kb is relatively short when the Pacbio long read sequencing technology has been adopted in genome sequencing. The authors stated they used a hybrid strategy for assemble, but only the MaSuRCA genome assembler is mentioned in the manuscript. The hybrid assemble strategy should be described in more detail. I wonder if other assemble algorithms have been implemented in current genome assemblies.

AU: We agree that there is still room from improvement, but not without adding more sequencing data. We tried different assembly strategies as it is now described in the methods section and this is the best we could obtain with the available sequence data. An assembly strategy using only the long reads with a subsequent polishing step with Illumina reads was not possible due to the relatively low coverage of PacBio sequences (<50X). The "hybrid assemble strategy" is referred only to the Masurca software that combines the benefits of deBruijn graph (for short reads) and Overlap-Layout-Consensus (for long reads) assembly approaches. Unfortunately, the PACBIO technology used was not the most recent one, this might partially explain the low N50. However, we consider that the assembly is of sufficient quality to carry out all the comparative analyses that represent the main point of the paper. Specific corrective actions have been taken to avoid that assembly fragmentation inflated the number of duplicated genes.

2 issues on the transcriptome comparison:

The authors obtained the *C. myersi* skeletal muscle transcriptome and compared to muscle transcriptomes of zebrafish, Tilapia and stickleback in the attempts to address the genomic basis for the icefish-specific increase of mitochondrial density and other physiological adaptations. However this comparison is far from convincing due to lack of samples from red-blooded Antarctic notothenioids and due to the limited tissue types (only one tissue) incorporated in the comparison.

*AU: We agree that red-blooded notothenioids are missing in the comparative transcriptome analysis. On the other hand, RNA-seq data for notothenioid species are relatively limited in public repositories and there are not sufficient biological replicates of the muscle tissue for a single species. To obtain novel RNA-seq data for a red-blooded species would require a new sampling campaign, which is extremely difficult before the next year. However, we propose a possible solution. We downloaded four biological replicates of RNA-seq data from three Antarctic red-blooded notothenioids (*P. charcoti*, *N. coriiceps*, *D. mawsoni*) from NCBI-SRA. We considered them altogether as representative of the red-blooded notothenioid white muscle transcriptome. This approach is potentially overly conservative because sampling across species might inflate the variance within one specific group (red-blooded notothenioids) since the other groups (zebrafish, tilapia, stickleback, icefish) represent within species variation. However, the obtained results are quite reassuring. The PCA clearly shows that icefish samples still cluster together. Likewise, red-blooded notothenioids also are closely grouped as are the non-Antarctic species. The results of differential expression analysis, especially the GSEA, but also gene-by-gene inspection, fully confirm that the icefish muscle transcriptome shows a mitochondrial-specific pattern, even when the comparison includes the closest red-blooded relatives, which live in the same subzero environment. All these new results are now incorporated in the revised version.*

With regard to the fact that the analysis is limited to a single tissue, it should be noted that we were testing a very specific hypothesis, i.e. the role of mitochondria in icefish adaptation to the hemoglobinless condition. Therefore, the choice of the muscle as a test tissue is not random. Moreover, muscle represents a

large portion of the fish body and any adaptive modification found in this tissue is likely extremely relevant for the species. We are convinced that other tissues might reveal other potential adaptations to the lack of haemoglobin, but a single paper cannot cover such a broad topic.

For example, elevation of GPATs and lipid synthesis in the icefish muscle transcriptome observed by the authors might merely be resultant of cold environment, rather than icefish-specific adaptation to their hemoglobinless condition. Alterations in lipid synthesis is a reported adaptation in other red-blooded notothenioid species (The genomic basis for colonizing the freezing Southern Ocean revealed by Antarctic toothfish and Patagonia robalo genomes. *Gigascience*. doi: 10.1093/gigascience/giz016, 2019).

AU: We agree. The results from the updated comparative transcriptomic analysis confirmed that a vast transcriptional program for "mitochondrial" proteins is specific to the hemoglobinless icefish also when compared to red-blooded notothenioids. However, when the genes encoding proteins involved in lipid metabolism are evaluated the evidence is less strong than in the case of mitochondrial shape and biogenesis. We have now discussed this point including the reference suggested by reviewer 1.

Authors also linked increased NOS1 transcripts to possible increase of NO levels in the icefish. This again lacks solid evidence when only one tissue is surveyed and the protein level of the enzyme is not assayed. I suggest the authors include the transcriptomes of other red-blooded Antarctic notothenioids and expand the comparison to other tissues.

AU: We agree. The topic of NO is very important for icefish evolution, but data from the present paper are not sufficient. This part was completely deleted.

3. Issues in gene duplication:

The authors compared the *C. myersi* Orthologous groups to other teleosts and drew conclusion that gene duplication might have contributed to mitochondrial biogenesis. Recently, three more notothenioid genomes including one icefish have been published (Antarctic blackfin icefish genome reveals adaptations to extreme environments. *Nat Ecol Evol*. 3(3):469-478, 2019; The genomic basis for colonizing the freezing Southern Ocean revealed by Antarctic toothfish and Patagonia robalo genomes. *Gigascience*. doi: 10.1093/gigascience/giz016, 2019). The authors should include these species in gene duplication analysis. Including of these species is necessary to address the genomic basis of icefish-specific phenotypes as the authors attempted throughout this manuscript.

*AU: We agree. The duplication analysis has been completely revised, first including all the novel genomic data (Kim et al. 2019, Chen et al. 2019) as suggested. Secondly, the potential inflation of duplicated genes due to genome fragmentation has been corrected as suggested by reviewer 2, considering only coding genes covering at least 60% of the full length sequence of the orthologous copy in the closest high quality genome (stickleback). Subsequently, we tested only whether duplicated genes are significantly enriched in mitochondrial genes by direct comparison of the mean copy number between two icefish species (*C. aceratus* and *C. myersi*) and the mean copy number between two Antarctic red-blooded notothenioids (*D. mawsoni* and *N. coriiceps*). The current analysis should be more robust for several reasons. It is based on two species per group and is performed across species that have all been corrected in the same way (see above). Secondly, enrichment analysis should be now relatively independent from species-specific biases (e.g. inflation of copy number) as it compares ratios within the same species (duplicated mitochondrial genes/non duplicated mitochondrial genes vs duplicated NOT-mitochondrial genes/non duplicated NOT-mitochondrial genes). The obtained results are stronger than the previous ones clearly suggesting that duplication of mitochondrial genes is one component of genomic adaptation to the hemoglobinless condition.*

Reviewer #2 (Remarks to the Author):

Dear authors,

After reading through your manuscript (COMMSBIO-19-0148-T) entitled “Comparative analysis of the icefish genome and transcriptome reveals the key role of mitochondria for a life without hemoglobin at subzero temperatures” my recommendation to the Editor is to accept with major revision.

First, I want to highlight that I think this is a nice piece of work that would be a nice addition to the already published literature on the icefish genetics/genomics in relation to adaptation to freezing temperatures (e.g. Bo-Mi et al. Nat Ecol and Evol 2019, Baalsrud et al. MBE 2018). In this regard, I would strongly urge the authors to take a closer look at this recently published paper in Nat Ecol and Evol, and refer to this work where appropriate.

The paper is well written and the analyses conducted are adequate for the questions in mind. However, I have some questions regarding the presentations of the analyses conducted as well as some suggestive additional analyses they should look into before the authors resubmit the ms, which I think will improve the paper further.

First, I have no concern regarding the genome assemblies. Think, that from the sequence data generated – they have produced OK draft genomes that could be used for detection of gene losses and gene expansions. However, it could have been nice with a table in the Supp – where some comparisons with other published genomes were highlighted. The *C. myersi* genome assembly is not that bad compared to many other teleost genomes out there.

AU: Thanks for the suggestion. We have now listed in a table the “quality” of all the recently published Antarctic fish genomes in terms of size, and BUSCO statistics.

But that said, the number of identified genes is in the upper segment (i.e. 38,127 putative protein-coding genes), and thus, for some of the analyses conducted within this ms caution should be made. Especially in regard to the gene duplication analysis.

Earlier reports have shown that differences in genome assembly qualities would impact the estimation of copy number of large gene families (e.g. Tørresen et al. BMC Genomics 2017). *AU: We agree. The other analyses (comparative transcriptomics, promoter analysis) should not be affected by the potential inflation in the estimate of copy number as they are performed considering orthogroups as comparison elements or one-to-one orthologs. However, it is certainly true that any duplication analysis might be biased in consequence of the potentially inflated number of gene copies due to fragmented assemblies. We decided to follow the reviewer’s suggestion and to “correct” the estimated number of gene copies. The followed approach was to use the phylogenetically closest species with a well annotated draft genome (stickleback) as a reference. Predicted coding genes in the genome of all five notothenioid species (*E. maclovinus*, *D. mawsoni*, *N. coriiceps*, *C. aceratus*, *C. myersi*) were blasted against the reference proteome of stickleback. Only those genes showing a significant match covering at least 60% of the corresponding protein in stickleback were retained for the duplication analysis. Although this approach is likely very conservative, it should significantly reduce the risk described by reviewer 2, i.e. counting two fragments of the same gene as two copies. Furthermore, we applied the same filter to all species included in the analysis (Antarctic notothenioids, either white- or red-blooded), therefore major biases should be excluded. This should also reply to the other reviewer’s points on the duplication issue. The current analysis*

should be more robust for several reasons. It is based on two species per group and is performed across species that have all been corrected in the same way (see above). Finally, the enrichment analysis should be relatively independent from species-specific biases (e.g. inflation of copy number) as it compares ratios within the same species (duplicated mitochondrial genes/non duplicated mitochondrial genes vs duplicated NOT-mitochondrial genes/non duplicated NOT-mitochondrial genes). The obtained results are stronger than the previous ones clearly suggesting that duplication of mitochondrial genes is one component of genomic adaptation to the hemoglobinless condition.

In light of the high number of duplicated genes, it would be nice with some more validations of that number. For instance, you could set up some criteria for the genes that are included in the final annotation:

1) Genes predicted only by ab-initio programs: these genes were considered good only if confirmed by at least four different ab-initio programs, if they were complete (with a start and a stop codon) and longer more than 300 base pairs.

2) Gene supported only by external evidence (e.g. proteins/RNA-seq): they need to be confirmed by at least two different evidence or by one external evidence and at least three different ab-initio gene predictors.

3) Predicted genes with a low ab-initio support (filter described at step 1) were further processed. Genes supported by less than four ab-initio programs were search against a database of teleost protein sequences. Proteins with a sequence coverage match higher than 70% and an e-value lower than $1E-20$ were recovered.”

None of these criteria guarantee that the genes are full-length, but criteria 1 and 2 should give long gene models. Could you validate the number of gene duplicates in some way? For instance, reporting the number of genes that have more than 70 % (80 % and 90 % maybe also) sequence coverage match against that database of teleost protein sequences. There might be other ways too, but the important factor is to avoid inflating the number by reporting parts of genes (which could happen, even with your criteria). You could have situations where you report the first 45 % of a gene as one gene, and the next 55 % as another. AU: *See above for a detailed explanation.*

Second, for the reconstruction of the LA and MN clusters in the different species (included some low coverage genomes already published) I see the need for a more detailed description both in the results section as well as in the methods. Especially since – as the authors state – some of the genomes used are highly fragmented. How the authors extracted the contigs from the different genome assemblies (how many unitigs/contigs in total was needed to do the reconstructions for the different species, and would be good to see the making of the clusters in the drawings where you f. ex. show the overlapping contigs and where they do not overlap, where you have the gaps). Further, an additional Supp table with the statistics re the reconstructions for the different genome assemblies linked to their publication (in addition to the added info in Fig 1 and in the text (as described above)) would be a nice add on as well.

As is, this information is totally lacking (from the methods/results and Supp).

AU: *Thanks for the suggestion. We have now repeated the analysis using the annotated genome assemblies for different notothenioid species. The previous results are fully confirmed. Indeed, the number of globin genes in the MN cluster in two well annotated red-blooded notothenioid genomes (*E. maclovinus*, *D. mawsoni*, *C. aceratus*) is higher than previously reported.*

Additionally, the fact that the authors used the genomic organization of the LA and MN clusters as given by

Opazo et al 2013, it would have been great to see them discuss if this could give rise to conflicting results or if they were able to identify the gene order within the two different gene clusters for the different species with high confidence (based on the contigs/unitigs extracted from the genome assemblies). These additions to the paper, would be highly valuable for the reliability/reproducibility of the methods/analyses conducted as well as a good way of showing others how to make use of these low coverage genomes.

AU: *Since we have now used higher quality annotated genomes, we believe this comment is no longer valid.*

And one last comment regarding this issue: Maybe the authors here should take a closer look at the recently published blackfin icefish genome in *Nat Ecol and Evol.* (also mentioned above to further confirm that their method is reliable).

AU: *Thanks for the suggestion. What we report for the second icefish sequenced genome is in full agreement with what reported for the blackfin icefish (Kim et al. 2019). In fact, the analysis of the globin gene clusters is much more detailed in our paper. We also provide an entirely different point of view on the icefish evolution. Finally, we now included the annotated genomes of two red-blooded notothenioids and RNA-seq data representative of Antarctic species.*

Third, I want to point out minor issues regarding the transcriptome analyses conducted. I have no comments on the analyses themselves, but I would have appreciated some more details around the sequencing data downloaded for the three species (*D. rerio*, *O. niloticus* and *G. aculeatus*). As is, the information given in the Supp table is scarce – here I would like to see information about the sequencing technology used (e.g HiSeq500 vs HiSeq4000 or maybe even HiSeq2000?), number of reads, number of replicates and so on for the different species.

Why? The main reason for this, is that it is shown that you easily detect differences between runs/library preps (and especially between different technologies) – that are then due to the runs/technologies and not differences due to biology. Even if I trust the differences found/reported here are due to difference in biology – I think this is something the authors could bring up in the discussion.

AU: *We agree. We report these data now in Supplementary Information. It should be noted that nearly all across species analyses reported so far partially rely on published data. There is unfortunately little room for improvement. However, we are quite confident that the RNA-seq comparative analysis is correct since also the addition of another group that pools four biological replicates of RNA-seq data from three red-blooded notothenioids (white muscle) did not alter the results (see the new PCA and detailed GSEA analysis)..*

Further, I find it highly noteworthy that they did the differentially expression analyses both as a comparison between icefish and all non-Antarctic species (based on the separation on the first component (PC1) as well as only using one of them (*G. aculeatus*). The similarity in results clearly demonstrate that pooling of the three non-Antarctic species is scientifically valid (since you actually see that there are some differences between these species given on the second component (PC2). So, for this, I would encourage the authors to summarize these results in the Supp as a nice validation of the comparison given in the main text (as I can see this information is not included in the current version of the ms or the Supp).

AU: *Reviewer 1 pointed out that a more meaningful analysis should have included red-blooded Antarctic species, in order to exclude that differences in gene expression between non-Antarctic species and icefish are related to the hemoglobinless condition rather than to the adaptation to the subzero environment. To answer this point, as already mentioned, additional RNA-seq data were included (from red-blooded Antarctic fish). To obtain novel RNA-seq data for a red-blooded species would have required a new sampling campaign, which is extremely difficult before the next year. However, we downloaded four biological*

replicates of RNA-seq data from three Antarctic red-blooded notothenioids (P. charcoti, N. coriiceps, D. mawsoni) from NCBI-SRA. We considered them altogether as representative of the red-blooded notothenioid white muscle transcriptome. This approach is potentially overly conservative because sampling across species might inflate the variance within one specific group (red-blooded notothenioids) since the other groups (zebrafish, tilapia, stickleback, icefish) represent within species variation. However, the obtained results are quite reassuring. The PCA clearly shows that icefish samples still cluster together. Likewise, red-blooded notothenioids also are closely grouped as are the non-Antarctic species. The results of differential expression analysis, especially the GSEA, but also gene-by-gene inspection, fully confirmed that the icefish muscle transcriptome shows a mitochondrial-specific pattern, even when the comparison includes the closest red-blooded relatives, which live in the same subzero environment. All these new results are now incorporated in the revised version.

Then to some of the results that could be looked further into. F. ex. the authors state in line 140-141: “Unlike the fate of the two globin clusters, inspection of the *C. myersi* genome revealed that a full-length copy is retained for all genes encoding key erythropoietic transcription factors that were found to be suppressed (full list in Table S2 of Xu et al. 2015²⁵) in the icefish (data not shown)”.

Could it be that the downregulation of these genes (due to living in freezing environments and no need for erythropoietic cells) further gave rise to the relaxed purifying selection on the Hb regions (as the authors here indicate with sequence decay). In other words what happened first? Maybe gene expression data from the red-blooded notothenioids, would be of added value?

AU: We agree that this is a very interesting biological question. However, we are afraid that it might not be easy to tackle. In evolutionary biology, the question about what came first is often elusive. In a partial attempt to add more evidence we performed a dedicated analysis on a set of seven key genes involved in erythropoiesis, which were found to be down-regulated in icefish hematopoietic tissues. A specific test for relaxed selection (RELAX, Whertheim et al. 2015) was applied, but the analysis of either single-gene analysis or a concatenated data set showed significant evidence of relaxed selection. This is very preliminary as few species could be used and a limited number of genes. However, it seems that the two globin gene clusters were subjected to a different set of evolutionary constraints than other genes important for erythropoiesis. We have included such novel evidence in the revised ms.

Similar, the result discussed in line 316-325:

“The transcription factor NRF2/GABPA is known to regulate genes involved in several processes including mitochondrial biogenesis^{61,62}. We found that in the *Chionodraco* genome the number of NRF2/GABPA binding sites was significantly higher in the upstream genomic region of genes that are overexpressed in muscle compared to the orthologous region in non-Antarctic fish genomes. It has been shown that multiplicity of transcription factor binding sites in the promoter region generally increases gene expression, with a clear positive relationship between the number of sites and average expression^{63,64}. This represents a second, much stronger evidence that the icefish genome has evolved structural modifications that are associated with major changes in muscle transcriptome profile. Therefore, genomic divergence seems at least partially responsible for the unique cellular adaptations that allow the hemoglobinless icefish to survive”.

Here, number of binding sites for these above-mentioned genes could be easily looked into for the red-blooded notothenioids (that they do have data for), which could give added value on the speculations if these genetic signatures are specific for the icefishes or not.

AU: *we agree. We have added to the comparison with non-Antarctic species, a similar test between C. myersi (white-blooded) and D. mawsoni (red-blooded). At variance with the analysis of duplicated genes where all available, good-quality notothenioid genomes were included, here a GFF file was required to determine the upstream genomic region. Such a file was not provided by the authors of the paper on C. aceratus. For N. coriiceps, the high fragmented status of the assembly also suggested against its inclusion in the promoter analysis. Nevertheless, the results of the two analyses are very consistent and confirm that the number of NRF2/GABPA binding sites is associated with higher gene expression even when the comparison is between red- and white-blooded notothenioids.*

Lastly, I would also recommend the authors to modify the title (since they seemingly here refer to one icefish genome? – but have in fact generated two – and used additional genome information from already published genomes). The title is:

“Comparative analysis of the icefish genome and transcriptome reveals the key role of mitochondria for a life without hemoglobin at subzero temperatures”

Would suggest to change to:

“Draft icefish genome assemblies and transcriptome data reveal key role of mitochondria for a life without hemoglobin at subzero temperatures”

AU: *we agree. The title has been changed accordingly, apart from the fact that we mention a single icefish genome in the title as report on the second Chionodraco species genome is only limited to Suppl. Info.*

Some additional more detailed comments that should be taken under considerations:

Line 90: “Through the analysis of the first draft genome of two white-blooded channichthyid species,.....

Should be re-written to:

“Through comparative genomic analyses of draft genome assemblies of two white-blooded channichthyid species,.....

See that you throughout the paper often refer to “comparative analysis” – I guess you did multiple analysis – and should do this quote in plural: “comparative analyses”. And preferably also add genomic, i.e. comparative genomic analyses.

AU: *we corrected this.*

Line 283-285:

“.....It was previously supposed that a single globin locus was present in notothenioids which was almost completely deleted in a single event in the common ancestor of channichthyids, possibly driving the evolution of the hemoglobinless condition²²”

Think this sentence is a bit vague – would suggest to re-write:

“It has been suggested that the notothenioids only harbor one globin locus – compared to other teleosts containing two – and that this locus was almost completely lost in a single event in the common ancestor of channichthyids, possibly driving the evolution of the hemoglobinless condition”

AU: *we agree. The sentence was changed as suggested.*

Then for line 287-289: It should be made clear that you here refer to your own results/data.

AU: *we agree. We have now added that the reconstruction of the evolution of the two globin gene clusters is reported here for the first time*

Line 415-417: This sentence gives very little information – more details should be given here and in the result section. And the sentence as such is poorly written:

AU: *we are not sure to understand. The sentence simply refers to genome data availability. We do not see how to improve it.*

“in *C. myersi*, *C. hamatus*, *N. coriiceps*, *P. charcoti*, *C. aceratus* and *E. maclovinus* genomes following the genomic organization described by Opazo and colleagues²⁴”.

Here you should describe in more detail how you did your BLAST search (e-value threshold) and also in more detail which flanking genes (the genomic organization given by Opazo et al).

AU: *We have provided more details about the BLAST search and the flanking regions used for characterize the two globin clusters MN and LA.*

Reviewer #3 (Remarks to the Author):

This study entitled "Comparative analysis of the icefish genome and transcriptome reveals the key role of mitochondria for a life without hemoglobin at subzero temperatures" presents new genome assemblies for two Antarctic icefish species as well as gene expression data for one of the two species to investigate the evolutionary loss of hemoglobin in these species. As this loss is a unique feature of icefishes among all vertebrates and furthermore one that may appear detrimental to the species but nevertheless is found in a rapidly radiating clade, the insights into its emergence are of great interest to evolutionary biologists. In particular, the results of this study show that the loss did not occur as a single event, but proceeded in multiple steps. This work thus lays an important foundation for future studies, that could employ genomic data of further icefish species to elucidate the details of this step-wise loss.

While I am not an expert on some of the applied workflows (e.g. the gene-expression comparisons), all of them appear to be done in an expertly manner (and some were established in a recent paper in *Nature Communications Biology*). The manuscript is well-written, but some details could be clarified as I will specify below. Besides these details, there is only one part of the study that I find somewhat disappointing, namely the contiguity of the genome assemblies. The genome assemblies of *Chionodraco myersi* and *C. hamatus* are based on extremely high coverage of 80-100x 150 bp Illumina reads (plus PacBio for *C. myersi*) and the genome sizes of the two species are not extreme (0.8-1.1 Gb), but nevertheless, the assemblies are quite fragmented. The one for *C. hamatus*, based on 80x Illumina data, has a N50 contig length of 2700 bp. In comparison, Malmstrøm et al. (<https://www.nature.com/articles/sdata2016132/tables/3>) sequenced the genome of the very closely related icefish species *Chaenocephalus aceratus* with a coverage of 36x 150 bp Illumina reads and obtained a contig N50 of 5500 bp. Moreover, the *C. hamatus* assembly presented in this study has nearly 40% missing data in scaffolds. I wonder if the assembly strategy with the CLC Genomics Workbench is not ideal and better assemblies could be obtained with other tools. I see no reason why the 80x data for *C. hamatus* should not yield a contiguity that is at least as good as that for *Chaenocephalus*

aceratus obtained by Malmstrøm et al. with 36x data. As the genome assemblies presented in this study will provide an important resource for the community, I urge the authors to also try other assembly strategies in order to optimize the assembly quality. Even though the assembly of *C. myersi* is not as fragmented as the one for *C. hamatus*, I would also expect that still better contiguity should be possible with the 100x Illumina read data plus PacBio reads, so I also suggest that the authors attempt to improve this assembly. At the very least, the choice of assembly tools should be justified in the manuscript, and particular reasons for the low contiguity should be discussed. That said, I note that the current low assembly contiguity apparently did not limit the permitted conclusions.

AU: *we agree. Following the suggestion of reviewer 1, we decided to report only the findings from C. myersi genome, which appears to have a less fragmented assembly. We tried different strategies to assemble the two genomes. It is possible that Chiodraco species, compared to Chaenocephalus aceratus have a higher heterozygosity and/or repetitive regions. In any event, we believe that the completeness of representation for coding genes is sufficient for the main purpose of the paper, which is to test a specific hypothesis on the role of mitochondrial in icefish.*

I also feel that the manuscript is currently a bit poor in figures. Better illustrations will attract more readers, thus, I suggest that at least a third figure is added. If images of the two species are available, these could be shown, other options include a map with the geographic ranges of the species, a more detailed image of the relationships and divergence times of the species, or some plots of the data that is currently presented only in tables in the main text and the supplement. All of these are just suggestions.

AU: *we agree. We have added a third figure describing the distribution of tau in different gene categories.*

Overall, I find this study valuable and interesting and think that it may become suitable for publication in Nature Communications Biology. A list of minor comments, in order of their appearance, is provided below.

Minor comments:

- l. 18; "vertebrate": I would use plural to clarify that there are multiple icefish species.

AU: *modified as suggested*

- l. 31: This sentence is slightly confusing because it sounds as if the family was first described by Ruud, but Ruud was only the first to describe their lack of hemoglobin.

AU: *modified as suggested*

- l. 40; "is not sufficient": It should be clarified for what this is not sufficient.

AU: *modified in "is not sufficient to compensate the lack of hemoglobin"*

- Results section "Icefish genome assembly and annotation": Might be worth repeating here what sequencing technology was used.

AU: *as the focus is on the evolution of the genome rather than the sequencing process we are convinced that it is sufficient to describe the sequencing technology in the methods section*

- l. 110: I suggest mentioning here which other notothenioid genomes were used.

AU: *we agree. We now refer to the papers reporting all the genomes being included in the analysis*

- l. 114; "notothenioids": Typo.

AU: *corrected*

- l. 116; "Bathidraconidae": Typo.

AU: *corrected*

- l. 120; "carried on": This should be "carried out on".

- AU: *corrected*

Figure 1: A reference should be given for the phylogeny.

AU: *done. There are now three relevant references for the tree*

The meaning of the asterisk, being the region displayed in B, should be mentioned. The meaning of the arrows in B should be explained in the figure legend, not just the main text.

AU: *done. Both asterisk and arrows are now explained in the figure legend.*

- l. 161-163: These expression differences do not necessarily have anything to do with hemoglobin loss. They could be related to many other differences between icefishes and the other taxa, including cold adaptation. I think this should be pointed out here or elsewhere.

AU: *we agree. However, as mentioned in our reply to a similar point raised by reviewer 1, we have now carried out the same comparative transcriptomic analysis against Antarctic red-blooded notothenioids, and results are fully consistent with those obtained against non-Antarctic species. We also discussed the point raised by both reviewers in the revised paper.*

- l. 164, "FDR", "NES": Abbreviations should be explained.

AU: *done. Now Fold Change (FC), False Discovery Rate (FDR), and Normalized Enrichment Score (NES) are explained when used for the first time in the revised text.*

- l. 170: It should be explained how these 16 genes were identified as being involved in mitoribosome assembly (presumably from the literature/databases). This also applies to the genes listed in Tables 2-4. It should be clarified if these lists are exhaustive or a selection.

AU: *these lists were identified from the literature and are as much as possible exhaustive, taking into account the fact that not all of them are found in all the analysed genomes/transcriptomes. References for these lists are included in the text.*

- l. 175: Period after "myersi" should be removed.

AU: *corrected*

- l. 178; "significant": The adverb should be used.

AU: *this sentence is not present in the revised version*

- l. 182: A reference is needed for the comment on the controversial role.

AU: *the reference was the one cited before (32). It has been now placed after the comment on the controversial role*

- l. 205; "among which": Word missing.

AU: *this sentence is not present in the revised version*

- l. 208: I don't think "justified" is the right word here.

AU: *this sentence is not present in the revised version*

- l. 210: I wouldn't say that this explains those phenotypes, but it is consistent with (or linked to) them.

AU: *this sentence is not present in the revised version*

- l. 217; "resulted overexpressed": Grammar not correct. "resulted" should be replaced, e.g. with "appeared".

AU: *corrected.*

- l. 220: The word "the" is missing before "that PGC1 α ".

AU: *corrected.*

- l. 228; "was showed": Should be "was shown".

AU: *corrected.*

- l. 236; "generates": Should be "generate".

AU: *this sentence is not present in the revised version*

- l. 240: Reference missing for the statement.

AU: *Four relevant references have been added*

- l. 247; "the first one": "of these" could be added.

AU: *corrected*

- l. 333: Accession numbers should be provided for the 18 genomes, as for some of these species multiple assemblies exist on NCBI and ENSEMBL.

AU: *we added NCBI and ENSEMBL accession codes for the 18 genomes.*

I. 335; "Protein dataset": Should be plural.

AU: *corrected*

- I. 341: Notothenia is not a member of Channichthyidae.

AU: *corrected*

- I. 351; "Both metrics...": This is of course not surprising given that the one is simply a fraction of the other and the distinction is only between positive and negative values. So, this should maybe be rephrased. And WMDelta is not provided anywhere it seems, but it could be added to Table S7.

AU: *we agree. Duplication analysis has been entirely revised and simplified. The only comparison is now against Antarctic red-blooded notothenioids, with a single metric, the "mean" difference (delta) between the mean number of gene copies in red-blooded notothenioids (N. coriiceps and D. mawsoni) and the mean number of copies in white-blooded notothenioids (C. myersi and C. aceratus).*

- I. 372; "Irt_CmyvsAll": This label is not reused, so it could be removed.

AU: *removed*

- I. 395, 396; "expression breath": Typo, should be "expression breadth".

AU: *corrected*

- I. 409; "arbitrary region": I'd rephrase. Not the region was arbitrary, just it's length.

AU: *we agree. Corrected as suggested.*

- I. 413; "C. myersi overexpressed": Grammar unclear, I'd rephrase as "OGs overexpressed in C. myersi".

AU: *corrected*

- I. 418; "blast" should be written in upper case.

AU: *corrected*

- I. 472: Names missing in reference.

AU: *added*

- Table 1: Abbreviations LSU/SSU (presumably large/small subunit) should be explained.

AU: *Table 1 is changed in revised manuscript*

- Supplementary Material, throughout: Version numbers should be given for all programs.

AU: *added version number for all programs used*

- Supplementary Material, page 3; "a fish protein database": Details about this database should be provided, also for databases mentioned on page 5.

AU: *added more details about databases used for genomes annotation.*

- Supplementary Material, all tables: Short table captions would be helpful.

AU: *added a short caption to all supplementary tables.*

- Supplementary Material, section "Gene prediction": There are lots of typos and grammar mistakes in this section. It should be proofread carefully.

AU: *corrected grammar and typos*

- Supplementary Material, Table S3: Rows with numbers of scaffolds/contigs > 10M nt can be deleted. And I don't understand how the shortest contig can have a length of 0.

AU: *corrected*

- Supplementary Material, Table S7: Information missing: What are the numbers in this table. Also abbreviations "GS", "OG_ID", "DeltaM" should be explained. And instead of "DeltaM" it should probably be "MDelta".

AU: *Table S7 concerning duplicated OGs has been deleted because it is no more necessary in the revised manuscript.*

- Supplementary Figure S1: Not sure where I am supposed to see the kmer-peaks indicated with dashed lines.

AU: *the dotted lines are those in vertical in the graphs and usually correspond to the predicted center of the peak. The big peaks at ± 90 (C. myersi) and ± 55 (C. hamatus) are the homozygous portion of the genomes.*

Reviewers' comments:

Reviewer #1 (Remarks to the Author):

The manuscript is improved to a large scale from the previous version. There are still a few issues remaining to be addressed:

1. There is a much larger number of protein coding genes predicted from *C. myersi* genome than the closely related *C. aceratus* (38140 vs. 30773). Such a big difference is unusual considering no extra whole genome duplication event has been suggested in the *C. myersi* genome. The nature and possible causes of this discrepancy should be addressed in the paper through examining the accuracy of the predicted gene models.
2. The authors indicated substantial gene duplications in *C. myersi*, however, there are no mentioning on exactly what genes are duplicated. It is highly desirable to present a list of the duplicated genes and their annotations.
3. The authors surveyed the expression level of duplicated genes versus non-duplicated genes and provided evidence that duplicated genes tend to be upregulated in mRNA expression and showed higher tissue specificity. However, the comparison could be problematic due to inaccurate estimation of gene copy numbers in some species. The two red-blooded notothenioid species (*D. mawsoni* and *N. coriiceps*) used in the comparison were sequenced by the short-read Illumina platform, which tend to miss out tandemly duplicated genes with high sequence similarity during genome assembly, and therefore the number of duplicated genes might be underestimated in these two species. Cautions needed to be taken when doing comparisons on the duplicated genes from genomes sequenced with different platforms.

Minor points:

1. It seems to be more appropriate to change the word 'icefish' in the title to "Chionodraco myersi".
2. Fig1B is a little confusing. The depicted region is shown to contain *nprt3* gene (the green box) and the *kank2* gene (the blue box) which are conserved between *E. maclovinus* and the two icefish in Fig 1A, but in Fig1B, the same genes are shown by a few fragmented light blue boxes denoted as "statistically significant blast matches". Are they the functional *nprt3* and *kank2* genes in the icefishes, or quickly diverged gene fragments? It's better consistent between Figs 1A and 1B.

Reviewer #2 (Remarks to the Author):

After reading through your revised manuscript (COMMSBIO-19-0148B) entitled "Draft icefish genome assembly and transcriptome data reveal the key role of mitochondria for a life without hemoglobin at subzero temperatures" my recommendation to the Editor is to accept with minor revision.

Overall, I think the authors have done a good job responding to the reviewers' comments. F. ex. by the inclusion of some of the recent published notothenioid genomes in their comparative genomic analyses as well as conducted additional analyses when asked to do so.

I think that the ms can be accepted for publication, but I have some minor comments that I hope the authors could pay some attention to before final submission.

Line 13:

Not sure why it only states "- including two other notothenioids -"

You have included two other icefishes (with blooded) is that what you mean? Then you should say that. Or state both red and white blooded - which is even more correct? Or at least the correct number of notothenioids used in this study - which is not only two?

"Comparative analyses of the icefish genome with those of other teleost species - including two other notothenioids -provided a novel perspective on the evolutionary loss of globin genes."

And PS you should enter a "space" after the last "-" before "provided."

Line 81-84:

"The role of gene duplication in Antarctic fish evolution has been further confirmed after the recent publication of the draft genomes of two red-blooded notothenioids, *Eleginops maclovinus* and *Dissostichus mawsoni* 30 and one white-blooded species, *Chaenocephalus aceratus* 31"

Punctum is lacking at the end of this sentence.

Line 117-119:

"*Eleginops maclovinus* is particularly interesting as it is the closest outgroup of all Antarctic notothenioids and it belongs to a lineage that never inhabited the waters surrounding Antarctica, while *P. charcoti* belongs to the family Bathydraconidae, which represents the closest living relatives to the hemoglobinless channichthyids 36,37."

Why do you here use the full Latin name here (has been fully listed on line 83)? And most importantly - think this information should be stated in the introduction not as part of the result section. In other words, would recommend to move up to end paragraph in the introduction (after line 84 where you have listed the species).

Line 146-148:

"Unlike the fate of the two globin clusters, inspection of the *C. myersi* genome revealed that a full-length copy is retained for all genes encoding key erythropoietic transcription factors that were found to be suppressed (full list in Table S2 of Xu et al. 201525) in the icefish (data not shown)".

Could preferably be re-written (as is it includes two "that" and is a bit unclear to me at least) to:

"Unlike the fate of the two globin clusters, full-length key erythropoietic transcription factors were identified (full list in Table S2 of Xu et al. 201525) in the *C. myersi* genome, but were found to be suppressed in the icefish (data not shown)".

Line 256:

"Massive gene duplication was reported to have affected the evolution of the notothenioid genome 27,29-31."

Would have said "is" not "was" - since you refer to already published and reported results:

"Massive gene duplication is reported to have affected the evolution of the notothenioid genome 27,29-31."

Line 268:

Same comment as above I guess:

...."effect 27,28, although experimental evidence was limited."

Suggest to re-write to:

...."effect 27,28, although experimental evidence is limited."

Line 396-399:

"In order to reduce the fragmentation of Antarctic genome assemblies, all protein-coding genes of *C. myersi*, *D. mawsoni*, *N. coriiceps*, *C. aceratus*, and *E. maclovinus* were compared to the proteome of *G. aculeatus* (Ensembl BROAD S1) by BLASTP analysis. Only proteins covering at least 60% of the orthologous sequence in stickleback were retained in the final datasets (Supplementary Table S4)."

I am really happy to see that the authors performed this analysis – and would recommend the authors to also list the "new" number of genes in the text (not only refer to the Supp Table S4). Think the number identified could be given in the result section re the "Comparative analysis of gene duplication", in Line 262.

And then at the end I have two final remarks:

- 1) Think that the authors should go over the ms one more time and see that they use the wording "the icefish" correctly.....throughout the ms they use the term when describing their species (f. ex Line 148). But should be careful since there are more than one icefish (and also more than one used in some of the analyses conducted here as well). Sometimes I feel the plural form would be more appropriate (f. ex. Line 112 and 115). So, I urge the authors to have one final go on using this term correctly.
- 2) And lastly, would also recommend the authors to proof read the ms one final time (maybe get one that is not that familiar with the ms to read it carefully) since there are some sentences in between that could benefit from some polishing.

Reviewer #3 (Remarks to the Author):

I thank the authors for addressing my earlier comments as well as those of the other reviewers. In my view the manuscript could be acceptable for publication after minor changes.

Minor comments:

I would find it helpful if you would use "icefishes" when referring to multiple icefish species and "icefish" when referring to a single species.

The manuscript contains both American and British English ("harbor" and "analysed"). Please make sure to use consistent spelling.

l. 15: You may want to rephrase that sentence. In its current form, it means that you compared with "temperate notothenioids" and with "red-blooded notothenioids".

l. 16: Typo, should be "channichthyids".

l. 17: "to generate" could be removed.

l. 35: Add "An" before "Increased".

l. 94: Replace "what" with "that".

l. 95: Remove "genome" after "icefish".

l. 101: Change words: "four other", or use "four more".

l. 111: Add "et al." after Opazo.

l. 117: Typo, should be "notothenioids".

l. 226: Typo in "notothenioids". I saw this typo multiple times in the newly written parts.

l. 226: Use "needs" or better "requires" instead of "need".

l. 238: Use hyphen in "up-regulated" and make sure that the spelling of "up-regulating" etc. is consistent throughout the manuscript.

l. 249-250: I'm not sure about the grammar here.

l. 257-258: Remove "nototheniod" before "genomes" and replace "ones" with "notothenioids". Note the correct spelling of "notothenioids".

l. 260: Remove comma after "species".

l. 266: Add "notothenioid" before "species".

l. 281: Use plural for "distributions".

l. 351: Replace "such" with "this"

l. 370: Replace "a second" with "additional". Like information, evidence can not be counted.

l. 371: Add "the" before "muscle".

l. 389: Add "and" before "Larimichthys"

l. 390-391: Sentence incomplete.

l. 391: Use upper case at beginning of sentence ("five").

l. 396-399: This does not make sense. The described BLASTP analysis does not reduce the fragmentation of the genome assemblies.

l. 402: Correct "blodded".

l. 403-404: Please clarify that you're referring to the mean NUMBERS of genes, not just the mean genes.

l. 420: Replace "on R" with "in R".

l. 421: Add equal sign between "n" and the number.

l. 448: Isn't that threshold for $M\Delta$ a bit low? So if an OG has just one more gene in either *C. myersi* or *C. aceratus* compared to *N. coriiceps* and *D. mawsoni*, the gene set is classified as duplicated? Maybe add "potentially" before "duplicated", given that this is followed up by the Fisher's exact test.

l. 461: This threshold for $M\Delta$ is inconsistent with the one specified above on line 448.

l. 466: Add "the" before "Jaspar".

l. 467: Add equal sign between "n" and number.

l. 468: Either rephrase to say that the length of 5 kb were arbitrarily chosen or just remove "of arbitrary length".

l. 479: Replace "well" with "high degree" or similar. An adjective is needed instead of an adverb.

l. 483: Add "the" before "NCBI database".

l. 483: Clarify what the number $10e-5$ is (the e-value I assume).

l. 485: Replace "UGENE software" with "the software UGENE".

l. 669: I assume something is missing after "logFC=".

Reply to reviewers' comments

Reviewer #1 (Remarks to the Author):

The manuscript is improved to a large scale from the previous version. There are still a few issues remaining to be addressed:

1. There is a much larger number of protein coding genes predicted from *C. myersi* genome than the closely related *C. aceratus* (38140 vs. 30773). Such a big difference is unusual considering no extra whole genome duplication event has been suggested in the *C. myersi* genome. The nature and possible causes of this discrepancy should be addressed in the paper through examining the accuracy of the predicted gene models.

Reply: We agree that the number of predicted coding genes is higher than the closely related species *C. aceratus*, but is rather difficult to fully understand the causes for such discrepancy. In fact, considering the number of genes involved (>7000), the analysis through manual curation of the annotation would be prohibitive. As the most likely cause for the higher number of duplicated genes in *C. myersi* is the fragmentation of the genome, we followed the suggestion of reviewer 2 and adopted a target approach to minimize the inflation of the number of duplicated genes in *C. myersi* and at the same time to increase the “comparability” across the four Antarctic fish genomes that are considered in the revised version of the manuscript. This analysis limited the divergence in the number of annotated genes between *C. myersi* and *C. aceratus* (26,800 and 24,900 respectively). While this confirms that assembly fragmentation is probably the major cause of the higher number of genes in *C. myersi*, inspection of the duplicated gene list shows that a large part of the remaining excess of annotated genes could be due to the expansion of few large gene families. For instance, just 15 orthogroups account for a difference of over 1,000 gene copies between *C. myersi* and *C. aceratus*. The largest ones of these gene families seem to encode for immune-related proteins (e.g. finTRIM, NRLP, Ig-like), which are known to be significantly duplicated in teleosts. In particular, finTRIMs are clustered into two orthogroups and appear to have nearly 300 copies in *C. myersi* and only 4 in *C. aceratus*. It is possible that our annotation pipeline and orthology analysis is incorrect, but when examining the number of gene copies in the same two orthogroups of other teleosts the number of copies is also quite high (e.g. tilapia has 84 copies, zebrafish 66). Large gene family expansions that occur independently in different lineages as a consequence of rapid gene birth-and-death process are frequent especially in immune-related genes. Because of the limited time to diverge, often different copies show small if any sequence differences, which makes difficult to distinguish between “alleles” and “duplicated copies”. We now explicitly discuss the discrepancy between *C. myersi* and *C. aceratus*, in the new revised version of the manuscript (lines 259-276). However, for all the analyses on duplicated genes we used the “corrected” list of duplicated genes, where differences between the two icefish species is much reduced. The novel data set that was generated across five notothenioid genomes is certainly not perfect, but it should provide a relatively reliable basis for comparative analysis.

2. The authors indicated substantial gene duplications in *C. myersi*, however, there are no mentioning on exactly what genes are duplicated. It is highly desirable to present a list of the duplicated genes and their annotations.

Reply: We agree on this suggestion. A list of the duplicated genes in *C. myersi* compared to other Antarctic species (the corrected data set) is now presented as supplementary info. It is too long to be included in the main text. Putative annotations are provided as well.

3. The authors surveyed the expression level of duplicated genes versus non-duplicated genes and provided evidence that duplicated genes tend to be upregulated in mRNA expression and showed higher tissue specificity. However, the comparison could be problematic due to inaccurate estimation of gene copy numbers in some species. The two red-blooded notothenioid species (*D. mawsoni* and *N. coriiceps*) used in the comparison were sequenced by the short-read Illumina platform, which tend to miss out tandemly duplicated genes with high sequence similarity during genome assembly, and therefore the number of duplicated genes might be underestimated in these two species. Cautions needed to be taken when doing comparisons on the duplicated genes from genomes sequenced with different platforms.

Reply: Yes, the comparison might be affected by the sequencing technology. However, the number of predicted genes after the “correction” (see point 1 above) appears to be similar across the five notothenioid species. Moreover, *D. mawsoni*, despite the use of only Illumina sequencing technology, seems to have the best completeness score. Finally, the reviewer’s comment is certainly correct, but it is valid for a very specific set of duplicated genes, those that have duplicated very recently, which we believe is a relatively small proportion of all orthogroups (see comment above). While we cannot completely rule out the effect of the different genome sequencing approaches, it should be noted that if this effect exists, there is no reason why it should affect more significantly the genes that are overexpressed in the muscle or that are more tissue specific. Therefore, the fact that we find for both categories a significant association with overexpression seems not to be attributable to a potential bias between sequencing technologies.

Minor points:

1. It seems to be more appropriate to change the word ‘icefish’ in the title to “*Chionodraco myersi*”.

Reply: We added the species name in the title.

2. Fig1B is a little confusing. The depicted region is shown to contain *npirt3* gene (the green box) and the *kank2* gene (the blue box) which are conserved between *E. maclovinus* and the two icefish in Fig 1A, but in Fig1B, the same genes are shown by a few fragmented light blue boxes denoted as “statistically significant blast matches”. Are they the functional *npirt3* and *kank2* genes in the icefishes, or quickly diverged gene fragments? It’s better consistent between Figs 1A and 1B.

Reply: The two genes are functional. Figure 1B shows the results of Blast searches, which do not show full similarity as they were carried out comparing nucleotide vs nucleotide sequences (BlastN), therefore similar regions are not continuous, especially considering intron and other non-coding regions within the genomic regions encoding *npirt3* and *kank2*. We have now clarified this in the legend to figure 1.

Reviewer #2 (Remarks to the Author):

After reading through your revised manuscript (COMMSBIO-19-0148B) entitled “Draft icefish genome assembly and transcriptome data reveal the key role of mitochondria for a life without hemoglobin at subzero temperatures” my recommendation to the Editor is to accept with minor revision.

Overall, I think the authors have done a good job responding to the reviewers’ comments. F. ex. by the inclusion of some of the recent published notothenioid genomes in their comparative genomic analyses as well as conducted additional analyses when asked to do so.

I think that the ms can be accepted for publication, but I have some minor comments that I hope the authors could pay some attention to before final submission.

Line 13:

Not sure why it only states “- including two other notothenioids –”

You have included two other icefishes (with blooded) is that what you mean? Then you should say that. Or state both red and white blooded – which is even more correct? Or at least the correct number of notothenioids used in this study – which is not only two?

“Comparative analyses of the icefish genome with those of other teleost species - including two other notothenioids -provided a novel perspective on the evolutionary loss of globin genes.”

Reply: We apologize, but this part was not corrected from the original version. Now the number of species mentioned should be correct.

And PS you should enter a “space” after the last “-“ before “provided.

Reply: Done.

Line 81-84:

“The role of gene duplication in Antarctic fish evolution has been further confirmed after the recent publication of the draft genomes of two red-blooded notothenioids, *Eleginops maclovinus* and *Dissostichus mawsoni* 30 and one white-blooded species, *Chaenocephalus aceratus* 31”

Punctum is lacking at the end of this sentence.

Reply: Corrected.

Line 117-119:

“Eleginops maclovinus is particularly interesting as it is the closest outgroup of all Antarctic notothenioids and it belongs to a lineage that never inhabited the waters surrounding Antarctica, while *P. charcoti* belongs to the family Bathydraconidae, which represents the closest living relatives to the hemoglobinless channichthyids 36, 37.”

Why do you here use the full Latin name here (has been fully listed on line 83)? And most importantly – think this information should be stated in the introduction not as part of the result section. In other words, would recommend to move up to end paragraph in the introduction (after line 84 where you have listed the species).

Reply: We agree. Now the sentence has been modified and moved to the introduction (lines 86-89) as suggested.

Line 146-148:

“Unlike the fate of the two globin clusters, inspection of the *C. myersi* genome revealed that a full-length copy is retained for all genes encoding key erythropoietic transcription factors that were found to be suppressed (full list in Table S2 of Xu et al. 201525) in the icefish (data not shown)”.

Could preferably be re-written (as is it includes two “that” and is a bit unclear to me at least) to:

“Unlike the fate of the two globin clusters, full-length key erythropoietic transcription factors were identified (full list in Table S2 of Xu et al. 201525) in the *C. myersi* genome, but were found to be suppressed in the icefish (data not shown)”.

Reply: The sentence has been modified as suggested.

Line 256:

“Massive gene duplication was reported to have affected the evolution of the notothenioid genome 27,29–31.”

Would have said “is” not “was” – since you refer to already published and reported results:

“Massive gene duplication is reported to have affected the evolution of the notothenioid genome 27,29–31.”

Reply: We agree. The sentence has been modified as suggested.

Line 268:

Same comment as above I guess:

....“effect 27,28, although experimental evidence was limited.”

Suggest to re-write to:

....“effect 27,28, although experimental evidence is limited.”

Reply: We agree. The sentence has been modified as suggested.

Line 396-399:

“In order to reduce the fragmentation of Antarctic genome assemblies, all protein-coding genes of *C. myersi*, *D. mawsoni*, *N. coriiceps*, *C. aceratus*, and *E. maclovinus* were compared to the proteome of *G. aculeatus* (Ensembl BROAD S1) by BLASTP analysis. Only proteins covering at least 60% of the orthologous sequence in stickleback were retained in the final datasets (Supplementary Table S4).”

I am really happy to see that the authors performed this analysis – and would recommend the authors to also list the “new” number of genes in the text (not only refer to the Supp Table S4). Think the number identified could be given in the result section re the “Comparative analysis of gene duplication”, in Line 262.

Reply: We agree. A new paragraph was added reporting the number of predicted coding genes in all four species before and after correction.

And then at the end I have two final remarks:

- 1) Think that the authors should go over the ms one more time and see that they use the wording “the icefish” correctly.....throughout the ms they use the term when describing their species (f. ex Line 148). But should be careful since there are more than one icefish (and also more than one used in some of the analyses conducted here as well). Sometimes I feel the plural form would be more appropriate (f. ex. Line 112 and 115). So, I urge the authors to have one final go on using this term correctly.

Reply: We agree. We have gone through the ms and substituted “icefish” with “*C. myersi*” when we were referring to just this specie, while using “icefish” where referring to more than one species.

- 2) And lastly, would also recommend the authors to proof read the ms one final time (maybe get one that is not that familiar with the ms to read it carefully) since there are some sentences in between that could benefit from some polishing.

Reply: We agree. We asked an English-speaking colleague who is not part of the authorship to read the ms and report any odd sentence.

Reviewer #3 (Remarks to the Author):

I thank the authors for addressing my earlier comments as well as those of the other reviewers. In my view the manuscript could be acceptable for publication after minor changes.

Minor comments:

I would find it helpful if you would use "icefishes" when referring to multiple icefish species and "icefish" when referring to a single species.

Reply: We understand that the plural of fish can be either "fish" or "fishes". This is why we used "icefish". Following the suggestion of reviewer 2, we have gone through the ms and substituted "icefish" with "C. myersi" when we were referring to just this specie, while using "icefish" where referring to more than one species.

The manuscript contains both American and British English ("harbor" and "analysed"). Please make sure to use consistent spelling.

Reply: Thanks. We have now uniformed the text to American English.

I. 15: You may want to rephrase that sentence. In its current form, it means that you compared with "temperate notothenioids" and with "red-blooded notothenioids".

Reply: Thanks. We have rephrased the sentence to avoid such a problem.

I. 16: Typo, should be "channichthyids".

Reply: Corrected.

I. 17: "to generate" could be removed.

Reply: Removed.

I. 35: Add "An" before "Increased".

Reply: Added.

I. 94: Replace "what" with "that".

Reply: Replaced.

I. 95: Remove "genome" after "icefish".

Reply: Removed.

l. 101: Change words: "four other", or use "four more".

Reply: Changed.

l. 111: Add "et al." after Opazo.

Reply: Added.

l. 117: Typo, should be "notothenioids".

Reply: Corrected.

l. 226: Typo in "notothenioids". I saw this typo multiple times in the newly written parts.

Reply: Checked throughout the ms and corrected.

l. 226: Use "needs" or better "requires" instead of "need".

Reply: Corrected to "requires".

l. 238: Use hyphen in "up-regulated" and make sure that the spelling of "up-regulating" etc. is consistent throughout the manuscript.

Reply: Checked throughout the ms and corrected.

l. 249-250: I'm not sure about the grammar here.

Reply: Corrected to "similar".

l. 257-258: Remove "nototheniod" before "genomes" and replace "ones" with "notothenioids". Note the correct spelling of "notothenioids".

Reply: Checked throughout the ms and corrected. The sentence was modified as suggested.

l. 260: Remove comma after "species".

Reply: Removed.

l. 266: Add "notothenioid" before "species".

Reply: Added.

l. 281: Use plural for "distributions".

Reply: Corrected.

l. 351: Replace "such" with "this"

Reply: Replaced.

l. 370: Replace "a second" with "additional". Like information, evidence can not be counted.

Reply: Replaced.

l. 371: Add "the" before "muscle".

Reply: Added.

l. 389: Add "and" before "Larimichthys"

Reply: Added.

l. 390-391: Sentence incomplete.

l. 391: Use upper case at beginning of sentence ("five").

Reply: A comma was needed at line 401 instead of a full stop. Now the sentence should not be incomplete.

l. 396-399: This does not make sense. The described BLASTP analysis does not reduce the fragmentation of the genome assemblies.

Reply: We agree. The BlastP analysis just reduces the inflation of duplicated genes due to assembly fragmentation. The sentence has been rephrased to clarify this.

l. 402: Correct "blodded".

Reply: Corrected.

l. 403-404: Please clarify that you're referring to the mean NUMBERS of genes, not just the mean genes.

Reply: Corrected.

l. 420: Replace "on R" with "in R".

Reply: Corrected.

l. 421: Add equal sign between "n" and the number.

Reply: Added.

l. 448: Isn't that threshold for $M\Delta$ a bit low? So if an OG has just one more gene in either *C. myersi* or *C. aceratus* compared to *N. coriiceps* and *D. mawsoni*, the gene set is classified as duplicated? Maybe add "potentially" before "duplicated", given that this is followed up by the Fisher's exact test.

Reply: We apologize. This sentence was not corrected from the original version. The correct threshold ($M\Delta \geq 1$) is not reported, which is now consistent with the next sentence.

l. 461: This threshold for $M\Delta$ is inconsistent with the one specified above on line 448.

Reply: See previous reply.

l. 466: Add "the" before "Jaspar".

Reply: Added.

l. 467: Add equal sign between "n" and number.

Reply: Added.

l. 468: Either rephrase to say that the length of 5 kb were arbitrarily chosen or just remove "of arbitrary length".

Reply: Corrected.

l. 479: Replace "well" with "high degree" or similar. An adjective is needed instead of an adverb.

Reply: Modified as suggested.

l. 483: Add "the" before "NCBI database".

Reply: Added.

l. 483: Clarify what the number $10e-5$ is (the e-value I assume).

Reply: Corrected.

l. 485: Replace "UGENE software" with "the software UGENE".

Reply: Replaced.

l. 669: I assume something is missing after "logFC=".

Reply: Yes, but the meaning of logFC was already explained earlier on in the main text, therefore we deleted "logFC=".

REVIEWERS' COMMENTS:

Reviewer #1 (Remarks to the Author):

The manuscript has been revised as suggested. I have no more comments. I recommend acceptance for publication.